# Robust Graph Attention for Graph Adversarial Attacks: An Information Bottleneck Inspired Approach

## Abstract

Graph Neural Networks (GNNs) have shown exceptional performance in learning node representations for node-level tasks such as node classification. However, traditional message-passing mechanisms solely based on graph structure in GNNs make them vulnerable to adversarial attacks. Attention-based GNNs have been utilized to improve the robustness of GNNs due to their capabilities to selectively emphasize informative signals over noisy or less relevant ones. However, existing works on robust graph attention methods do not realize the correlation between improved robustness and better adherence to the IB principle of attention-based GNNs. In this work, we find that the IB loss of attention-based GNNs is a strong indicator of their robustness against variant graph adversarial attacks. Attention-based GNNs with lower IB loss learn node representations that correlate less to the input training data while aligning better with the target outputs. Due to better adhering to the IB principle, attention-based GNNs with lower IB loss usually show stronger robustness against graph adversarial attacks. Inspired by such observation, we propose a novel graph attention method termed Robust Graph Attention inspired by Information Bottleneck, or RGA-IB, which explicitly minimizes the IB loss of a multi-layer GNN through a carefully designed graph attention mechanism. Extensive experiment results on semi-supervised node classification under variant graph adversarial attacks show that GNNs equipped with RGA-IB exhibit lower IB loss, which indicates better adherence to the IB principle, and show significantly improved node classification accuracy under graph adversarial attacks compared to existing robust GNNs. The code of RGA-IB is available at https://anonymous.4open.science/r/RGA-IB-A47F/.

## 1 Introduction

As generalizations of Deep Neural Networks (DNNs), Graph Neural Networks (GNNs) have emerged as popular tools for machine learning on graph-structured data (Kipf & Welling, 2017; Bruna et al., 2014; Hamilton et al., 2017; Xu et al., 2019b). Most prevailing GNNs (Kipf & Welling, 2017; Hamilton et al., 2017) follow the message-passing scheme and learn the representation of each node by iteratively transforming and propagating the information within its neighborhood. Benefiting from such merits, GNNs show dominant performance on various graph learning tasks, such as node classification (Ding et al., 2023), link prediction (Zhang & Chen, 2018), and graph classification (Zeng & Xie, 2020). Among different graph learning tasks, semi-supervised node classification, aiming at predicting the labels for a set of unlabeled nodes in a partially labeled attributed graph (Kipf & Welling, 2017), benefits the most from the message-passing scheme as it allows information from labeled nodes to propagate and influence the predictions for unlabeled nodes. However, the message-passing scheme also makes GNNs vulnerable to adversarial attacks (Zügner et al., 2018). Recent works (Zügner & Günnemann, 2019; Sun et al., 2020) have shown that by carefully perturbing only a small number of edges or nodes in the graph, adversarial attacks can catastrophically reduce the performance of GNNs in predicting the labels on either all unlabeled nodes (Zügner & Günnemann, 2019) or only a small targeted set of unlabeled nodes (Zügner et al., 2018) in the semi-supervised node classification task. Some efforts have been devoted to improving the robustness of GNNs by adversarial training (Feng et al., 2019; Li et al., 2022), graph pre-processing (Wu

et al., 2019; Entezari et al., 2020; Jin et al., 2020; Lei et al., 2022), and model robustification (Zhao et al., 2023; Song et al., 2022; Jia et al., 2023).

Adaptively assigning weights to the neighbors of a node in the message-passing scheme, graph attention modules (Veličković et al., 2018; Zhang & Zitnik, 2020; Yang et al., 2021b; Feng et al., 2021; Fountoulakis et al., 2023; Wu et al., 2023a) have recently drawn increasing attention in improving the robustness of GNNs among model robustification methods. Early graph attention methods, such as GAT (Veličković et al., 2018), assign weights to edges in the graph by either estimating the uncertainties of edges (Feng et al., 2021; Yang et al., 2021b) or the similarities between neighboring nodes (Veličković et al., 2018; Zhang & Zitnik, 2020). Recent advancements of Transformer-based GNNs (Wu et al., 2023b; Fountoulakis et al., 2023; Wu et al., 2023a) such as Difformer (Wu et al., 2023a) have introduced mechanisms that extend beyond immediate neighbors to capture the dense correlations among all the nodes in the graph. By focusing on the most relevant and reliable information within the graph structure, graph attention networks dynamically modulate the influence of the neighbors of each node, improving the resilience of the GNNs against malicious graph structures and features. For example, RGCN (Zhu et al., 2019) and GAR (Fountoulakis et al., 2023) design novel graph attention modules to improve the robustness of GNNs against graph adversarial attacks. However, the attention modules in existing attention-based GNNs are often empirically designed and lack theoretical support.

**Understanding Robust Graph Attention from the Information Bottleneck (IB) perspective.** In this work, we understand the robust graph attention mechanism from the Information Bottleneck (IB) principle. The IB principle (Tishby et al., 2000) encourages maximizing the mutual information between the node representation and input features while minimizing the mutual information between the node representation and class labels. Let $X$ be the random variable representing the input features and $Z$ be the random variable representing the node representations to be learned by the GNNs. Let $Y$ be the ground truth training labels for the node classification task. The IB principle is to maximize the mutual information between $Z$ and $Y$ while minimizing the mutual information between $Z$ and $X$, that is, the IB loss $\text{IB}(Z, X, Y) = I(Z, X) - I(Z, Y)$. Lower IB loss indicates better adherence to the IB principle. As a result, learned GNNs adhering to the IB principle naturally avoid overfitting to the inputs and become more robust to adversarial attacks in the input graph data. Recent works find that minimizing the IB loss can improve the adversarial robustness of DNNs (Wang et al., 2021; Kuang et al., 2023) and GNNs (Wu et al., 2020). We provide new insights into the underlying connection between the IB principle (Tishby et al., 2000) and the robustness of GNNs with graph attention modules which has not been revealed in the graph learning literature. The graph attention operation selectively aggregates informative signals over noisy or less relevant ones. Such selective focus leads to more relevant and compact node representations which correlate less to the input graph data while aligning better with the training class labels, thereby adhering better to the IB principle with a lower IB loss. This targeted refinement of node representations through the attention mechanism results in an efficient compression of information, which enhances the generalization capabilities of the GNNs with graph attention.

Existing methods do not realize the connection between the graph attention mechanism and the IB principle. As evidenced in Table 5, there is a strong correlation between the IB loss and node classification accuracy under adversarial attacks for graph attention methods. IB loss can be regarded as an indicator of the robustness of the graph attention models, as the graph attention methods showing better robustness usually feature lower IB loss. Motivated by such observations, we propose a novel graph attention method, Robust Graph Attention inspired by Information Bottleneck, or RGA-IB, to explicitly reduce IB loss of GNNs with carefully designed robust graph attention layers, termed the RGA-IB layers.

## 1.1 CONTRIBUTIONS

Our contributions are presented as follows.

First, we introduce a novel graph attention method termed Robust Graph Attention inspired by Information Bottleneck, or RGA-IB. RGA-IB is motivated by the connection between the principle of Information Bottleneck (IB) and the robustness of graph attention methods against adversarial attacks. The superior robustness of various GNNs with graph attention mechanisms, such as GAR (Fountoulakis et al., 2023), as shown in Table 5, can be explained by their better adherence to the IB principle evidenced by their lower IB loss compared to other graph attention methods. Although graph attention operation has been applied to improve the robustness of GNNs by se-

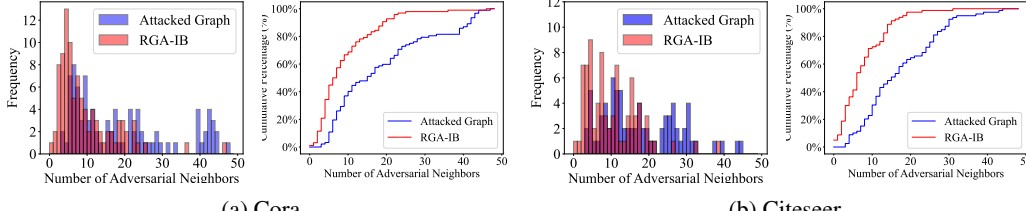

(a) Cora                                    (b) Citeseer

Figure 1: Comparisons on the frequency and cumulative frequency of the number of adversarial neighbors in the attacked graph and the RGA-IB attention graph on Cora and Citeseer. Nettack with an attack budget of $5$ is adopted for this experiment. For each node in the attacked graph or the RGA-IB attention graph, we count its adversarial neighbors which are the perturbed nodes within two hops of that node. This is because node representations in existing IB based works, including GIB (Wu et al., 2020), RG-GIB (Dai et al., 2023b), and UGRL (Wang et al., 2023b), are limited by a two-hop neighborhood. The RGA-IB attention graph is created such that two nodes are connected only when the attention weight between them is larger than $0.2$. The figures illustrate that most nodes of the RGA-IB attention graph have much fewer adversarial neighbors compared to the attacked graph. For example, more than $90\%$ of nodes in the RGA-IB attention graph have less than $20$ adversarial neighbors in Cora. In contrast, only $60\%$ of nodes in the attacked graph have less than $20$ adversarial neighbors in Cora. Such observations demonstrate that the dense graph attention in RGA-IB significantly mitigates the propagation of adversarial information on the attacked graph. Details on the calculation of the frequency and cumulative frequency of the number of adversarial neighbors in the attacked graph and the RGA-IB attention graph are deferred to Section C.2 of the appendix. Results on Pubmed and Polblogs are deferred to Figure 3 in Section C.2 of the appendix.

lectively capturing node-wise correlations, all existing robust graph attention methods (Feng et al., 2021; Zhu et al., 2019; Fountoulakis et al., 2023) do not explicitly reduce the IB loss. As shown in Table 5 in Section 4.3, attention-based GNNs with lower IB loss show improved robustness to adversarial attacks than attention-based GNNs with higher IB loss. To further reduce the IB loss, we propose RGA-IB, which explicitly minimizes the IB loss of a GNN through a carefully designed graph attention.

Second, to explicitly reduce the IB loss of a GNN with graph attention, we view the GNN with multiple RGA-IB layers as an iterative process for the reduction of the IB loss by gradient descent, and each RGA-IB layer simulates one-step gradient descent on the IB loss. Inspired by this understanding, the attention weight matrix at the current layer is generated from the attention weight matrix at the previous layers, and the input node features at the current layer, following the formula of gradient in Equation (1) in Section 3.2. As a result, the RGA-IB network with RGA-IB layers enjoys reduced IB loss compared to existing graph attention methods, which is evidenced in Table 5 in Section 4.3. As evidenced by results in Table 4 in Section 4.3, RGA-IB gradually reduces the IB loss to a lower level at deeper layers compared to existing graph attention methods. In addition, extensive evaluation results on public graph benchmarks for semi-supervised node classification under different categories of graph adversarial attacks in Section 4.2 demonstrate the effectiveness of explicitly reducing IB loss with RGA-IB for improving robustness.

It is worthwhile to mention that RGA-IB is significantly different from existing robust GNNs designed by the IB principle. GIB (Wu et al., 2020) proposes to learn minimal sufficient node representations for node classification under graph adversarial attacks by explicitly minimizing the variational upper bound of the IB loss, adhering closely to the IB principle. Although GIB shows improved robustness against graph adversarial attacks, the node representations learned by GIB are limited by a local dependency assumption where the representation of a node can only depend on the features of neighboring nodes within two hops. More recent superior GNNs such as Difformer (Wu et al., 2023b) find that node representations that capture all-pair node correlations beyond neighboring nodes demonstrate better performance for node-level learning tasks such as node classification. Following such observations, we design the RGA-IB layer, which learns node representations with dense graph attention to capture the correlation between all pairs of nodes in the graph for reducing the IB loss. Not limited by the local dependency assumption, the RGA-IB network demonstrates significantly better robustness against adversarial attacks for semi-supervised node classification. Experiment results in Table 8 in Section C.1 of the appendix compare RGA-IB with ablation model, RGA-IB$_{local}$, that only captures local node correlations further evidence that global correlation learn-

ing in RGA-IB is beneficial for reducing the IB loss of robust attention-based GNNs. Moreover, Figure 1 illustrates that the dense graph attention by RGA-IB considerably mitigates the propagation from the adversarial neighbors, which are the nodes perturbed by graph adversarial attack, to a target node. In addition, UGRL (Wang et al., 2023b) and RG-GIB (Dai et al., 2023a) also adopt the IB principle to improve the adversarial robustness of node representation learning. However, both UGRL and RG-GIB adopt the conventional neighborhood aggregation scheme of GNNs on the original input graph, thus exposing additional vulnerabilities to attacks on the graph structures. In contrast, RGA-IB reduces the IB loss by graph attention operation using dense node correlations, thus adaptively aggregating informative features from the potential new neighbors of a node which are not present in the given graph for learning robust node representation.

## 2 RELATED WORKS

### 2.1 GRAPH ADVERSARIAL ATTACKS AND DEFENSE

Despite the success of Graph Neural Networks (GNNs) in various applications (Kipf & Welling, 2017; Zhang & Chen, 2018) on the graph-structured data, recent works have shown that GNNs are vulnerable to adversarial attacks. Graph adversarial attacks (Dai et al., 2022) aim to degrade the performance of GNNs by injecting deliberate perturbations into the graph dataset. Based on whether the goal of the attacker is to reduce the performance of the GNN on a set of target instances or reduce the overall performance of the GNN model on the targeted datasets, threat models can also be categorized as: (1) targeted attack (Zügner et al., 2018), which aims to fool a GNN model to misclassify a set of target nodes, and (2) untargeted attack (Zügner & Günnemann, 2019; Sun et al., 2020), which aims to reduce the overall performance of the GNN model on the target dataset. To address the vulnerability of GNNs to adversarial attacks, various robust learning methods have been put forward, which can be categorized into three classes: Adversarial Training, Graph Processing, and Model Robustification. Adversarial training methods (Feng et al., 2019; Li et al., 2022) train robust models on a training set augmented with handcrafted adversarial samples. Graph processing methods (Wu et al., 2019; Entezari et al., 2020; Jin et al., 2020; Lei et al., 2022) aim to purify the graph data and remove adversarial perturbations. For example, Pro-GNN (Jin et al., 2020) learns a clean graph structure by preserving sparse and low-rank properties in the adjacency matrix, as well as feature smoothness during training. Model robustification methods (Xie et al., 2023; Chamberlain et al., 2021; Rusch et al., 2022; Song et al., 2022; Zhao et al., 2023; Jia et al., 2023) refine the GNN models to prepare against potential adversarial threats. For example, G-RNA (Xie et al., 2023) adopts graph neural architecture methods to search for robust architectures for GNNs. More recently, GCORNs (Abbahaddou et al., 2024) proposes to improve the robustness of GNNs against adversarial attacks by orthonormalization of the weight matrices.

### 2.2 ATTENTION-BASED GRAPH NEURAL NETWORKS

In the graph domain, Graph Attention Networks (GAT) (Veličković et al., 2018) firstly adopts an attention mechanism in designing GNNs and shows improved performance in node classification. In addition, GAT is found to be more robust to various types of graph adversarial attacks (Zügner & Günnemann, 2019; Sun et al., 2020) attributed to the capability of attention mechanism in learning robust representations (Goyal et al., 2023; Zhou et al., 2022). Following that, GNNGuard (Zhang & Zitnik, 2020) proposes a novel attention module that estimates neighbor importance based on the assumption that nodes with similar structural roles are more likely to interact than dissimilar nodes. TWIRLS (Yang et al., 2021b) introduces an attention mechanism that weights the edges in GNN with an energy function measuring the edge uncertainty. At the same time, UAG (Feng et al., 2021) also proposes an uncertainty-aware graph attention model that dynamically adjusts the impact of one node towards its neighboring nodes based on its Bayesian uncertainty. Following the design of self-attention modules in transformers (Vaswani et al., 2017), NodeFormer (Wu et al., 2022) explores layer-wise message passing over latent graphs potentially connecting all nodes in attention-based transformer networks. SGFormer (Wu et al., 2023b) proposes a simple yet effective attention-based transformer architecture to capture all-pair influence beyond neighboring nodes. Recently, GAR(Fountoulakis et al., 2023) proves that graph attention modules exhibit strictly better robustness against structural noise in the graph over both the graph convolution and linear classifier.

### 2.3 INFORMATION BOTTLENECK AND ITS APPLICATION FOR GNNS

The Information Bottleneck (IB) (Tishby et al., 2000) principle aims to learn latent representations of data that retain information relevant to the target task while minimizing redundant information from

the input. Deep VIB (Alemi et al., 2017) firstly introduces the IB principle as the objective for the training of deep neural networks. Inspired by the IB principle, (Lai et al., 2021) proposes a spatial attention module that minimizes the IB loss on the attention-modulated representation. Following that, (Zhou et al., 2022) proves that self-attention can be interpreted as an iterative optimization of the IB objective. To justify the benefits of minimizing IB loss in deep learning, recent works (Amjad & Geiger, 2020; Kawaguchi et al., 2023) theoretically prove that controlling IB loss is one way to control generalization errors in deep learning. In addition, recent works also find that (Wang et al., 2021; Kuang et al., 2023) minimizing the IB loss can improve the adversarial robustness of DNNs.

**Information Bottleneck for GNNs.** More recently, the IB principle has been successfully adapted to different graph learning tasks (Wu et al., 2020; Sun et al., 2022; Xu et al., 2021; Yu et al., 2022; Miao et al., 2022; Liu et al., 2023a; Dai et al., 2023a) to learn more representative and robust representations. Graph Information Bottleneck (GIB) (Wu et al., 2020) first extends the IB principle to learn adversarial robust node representations under a local dependency assumption where the representation of a node can only depend on the features of neighboring nodes within two hops. UGRL (Wang et al., 2023b) proposes to learn robust node representations against adversarial perturbations in unsupervised node classification. RG-GIB (Dai et al., 2023a) show that the IB principle can benefit both the membership privacy and adversarial robustness of GNNs by regularizing the predictions on labeled samples. InfoGCL (Xu et al., 2021) reduces the mutual information between contrastive parts while keeping task-relevant information for contrastive graph representation learning. HGIB (Yang et al., 2021a) adopts the IB principle for unsupervised representation learning on heterogeneous graphs. In addition, the IB principle is also widely used for the graph-level learning tasks such as graph classification (Yu et al., 2022; Miao et al., 2022; Sun et al., 2022; Seo et al., 2023; Wang et al., 2023a), graph-level anomaly detection (Liu et al., 2023b), graph reconstruction (Zhou et al., 2023), and graph condensation (Fang et al., 2024). For example, PGIB (Seo et al., 2023) incorporates prototype learning with the IB principle for explainable graph classification. VIB-GSL (Sun et al., 2022) learns sparse graph structures that are both informative and robust for graph classification guided by the IB principle. IB principle has also been employed to improve the performance of temporal GNNs. For instance, DGIB (Yuan et al., 2024) and TGIB (Seo et al., 2024) propose to incorporate the IB principle for temporal link prediction. However, both DGIB and TGIB suffer from the local dependency assumption as they adopt the same IB minimization framework as GIB (Wu et al., 2020). Our work focuses on studying the effectiveness of the IB principle for robust node classification. Therefore, we do not compare with works employing the IB principle for graph-level learning tasks and temporal graph learning tasks surveyed above in our experiments.

## 3 METHODS

In this section, we propose a novel graph attention module inspired by the Information Bottleneck (IB) principle, termed Robust Graph Attention (RGA-IB). In Section 3.1, we detail the notations of the attributed graph and introduce the formulation of RGA-IB. Next, we present how the attention weight matrix of RGA-IB is generated to reduce IB loss in Section 3.2.

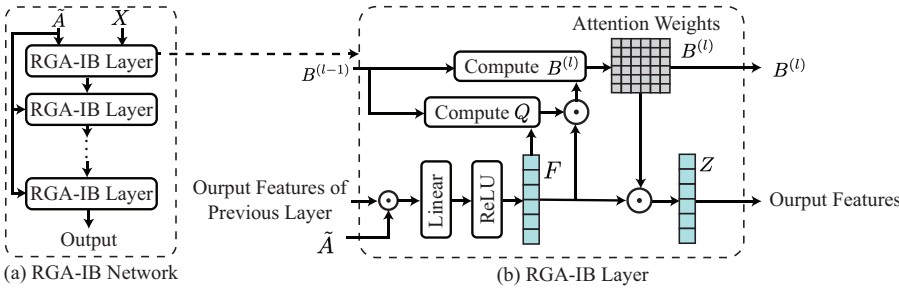

(a) RGA-IB Network    (b) RGA-IB Layer

Figure 2: Overall framework of GNN with multiple Robust Graph Attention (RGA-IB) layers and the detailed structure of the RGA-IB layer. An RGA-IB network generates the node representations given the normalized adjacency matrix $\tilde{A}$ and node features $X$. Given the output features of the previous layer, the attention weight matrix $B^{(\ell-1)}$ of the previous layer, and the normalized adjacency matrix $\tilde{A}$, an RGA-IB layer first computes the latent node features $F$. Next, the attention weight matrix $B^{(l)}$ is generated by Equation (1). After obtaining the attention weight matrix $B^{(\ell)}$, the attention augmented node features are computed $Z = B^{(l)}F$. Detailed formulation of RGA-IB can be found in Section 3.1.

### 3.1 Robust Graph Attention inspired by Information Bottleneck (RGA-IB)

We begin by formally defining the notations used for an attributed graph. Subsequently, we present the detailed formulation of Robust Graph Attention inspired by Information Bottleneck, or RGA-IB.

**Attributed Graph.** An attributed graph with $N$ nodes is formally denoted as $\mathcal{G} = (\mathcal{V}, X, A)$, where $\mathcal{V} = \{v_1, v_2, \ldots, v_N\}$ represents the nodes and $\mathcal{E} \subseteq \mathcal{V} \times \mathcal{V}$ represents the edges. The node attributes are represented by $X \in \mathbb{R}^{N \times D}$, where each row $X_i \in \mathbb{R}^D$ corresponds to the attributes of node $i$ and $D$ is the attribute dimension. The adjacency matrix $A \in \{0, 1\}^{N \times N}$ defines the connections in the graph $\mathcal{G}$, with $A_{ij} = 1$ if and only if there is an edge $(v_i, v_j) \in \mathcal{E}$. The adjacency matrix with self-loops included is given by $\widehat{A} = A + I$, and the corresponding diagonal degree matrix is $\widehat{D}$. $\tilde{A} = \widehat{D}^{-\frac{1}{2}} \widehat{A} \widehat{D}^{-\frac{1}{2}}$ is the normalized graph Laplacian.

**Robust Graph Attention inspired by Information Bottleneck (RGA-IB).** In this work, we aim to propose a novel graph attention operation, termed Robust Graph Attention inspired by Information Bottleneck (IB), or RGA-IB, which can be incorporated into multi-layer GNNs for semi-supervised node classification. We first introduce the setup of graph attention operation in GNNs. Let $X \in \mathbb{R}^{N \times D}$ be the input feature matrix to the graph attention operation. The output features of a GNN layer with graph attention operation are then calculated by $Z = B\sigma\left(\tilde{A}XW\right)$, where $W \in \mathbb{R}^{D \times D'}$ is the weight matrix for the linear transformation of the input. $D'$ is the hidden dimension of the linear transformation. $B \in \mathbb{R}^{N \times N}$ is the attention weight matrix, where $B_{ij}$ denotes the feature correlation between node $v_i$ and node $v_j$. $\sigma(\cdot)$ is a non-linear activation function such as ReLU. Let $F = \sigma(\tilde{A}XW) \in \mathbb{R}^{D'}$ be the latent node features before applying the graph attention operation. The output features of a GNN layer with graph attention operation can be denoted as $Z = BF$.

Graph attention operation has been widely studied for designing Graph Neural Networks (GNNs) that adaptively model the feature correlation between nodes in a graph (Lee et al., 2019; Veličković et al., 2018). The major differences between different graph attention operations lie in how the attention weight matrix $B$ is computed. For instance, GAT (Veličković et al., 2018) concatenates the node features of different nodes and applies a linear transformation to compute their similarity. Difformer (Wu et al., 2023a) adopts the dot-product operation widely used in the self-attention operation in transformers (Vaswani et al., 2017) to capture pair-wise similarities among nodes in a graph. Although the graph attention operation has been widely studied, all existing graph attention methods do not realize that the graph attention operation can reduce the IB loss of GNNs by enhancing the correlation of learned features with class labels while reducing their correlation with the input. The enhanced robustness of GNNs with graph attention is attributed to the capabilities of the graph attention modules to enhance informative signals while diminishing noise or less pertinent details. The selective attention mechanism produces node representations that are less correlated with the input training data, which might contain adversarial noises, and more aligned with the desired outputs, adhering more closely to the Information IB principle. Inspired by the observation that graph attention can reduce the IB loss of GNNs, we propose RGA-IB that explicitly reduces the IB loss via graph attention operations at consecutive layers. Similar to existing graph attention methods, a linear layer followed by a non-linear activation function is first applied to the input features at each RGA-IB layer to obtain the latent node features $F$. At the $\ell$-th RGA-IB layer in a multi-layer RGA-IB network, the output features are computed by $Z = B^{(\ell)}\sigma\left(\tilde{A}XW\right)$, where $B^{(\ell)}$ is the graph attention weight matrix of the $\ell$-th RGA-IB layer. As illustrated in Figure 2, the graph attention weight matrix $B^{(\ell)}$ of the $\ell$-th RGA-IB layer is generated by the attention weight matrix $B^{(\ell-1)}$ of the previous layer and the latent node features $F$, which is motivated by reducing the IB loss and detailed in Section 3.2. The attention weight matrix $B^{(1)}$ for the first layer of the RGA-IB network is generated from the input node feature $X$ with a GAT layer (Veličković et al., 2018).

### 3.2 Generating Attention Weight Matrix of RGA-IB by Reducing the IB Loss

In this section, we detail the process of generating the attention weight matrix for an RGA-IB layer inspired by reducing the Information Bottleneck (IB) loss. We first describe the configuration in which the IB loss for a GNN is specified.

Given the training data $\{X_i, y_i\}_{i=1}^N$, where $X_i \in \mathbb{R}^D$ is the $i$-th input node feature, and $y_i$ is the corresponding class label, we first specify how to compute the IB loss, $\text{IB}(Z, X, Y) = I(Z, X) - I(Z, Y)$, where $X$ is a random variable representing the input

features, which takes values in $\{X_i\}_{i=1}^N$. $Y$ is a random variable representing the class label, which takes values in $\{y_i\}_{i=1}^N$. $Z$ is a random variable representing the node representations, which takes values in $\{Z_i\}_{i=1}^N$ with $Z_i \in \mathbb{R}^D$ being the node representation of node $v_i$. $I(\cdot, \cdot)$ denotes the mutual information. To compute the mutual information, we first calculate the class centroids on $\{Z_i\}_{i=1}^N$ and $\{X_i\}_{i=1}^N$, resulting in class centroids $\{\mathcal{C}_a\}_{a=1}^C$ and $\left\{\mathcal{C}_b'\right\}_{b=1}^C$ for representation space and input feature space respectively, where $C$ is the number of classes. Then we define the probability that node representation $Z$ belongs to class $a$ as $\Pr[Z \in a] = \frac{1}{N} \sum_{i=1}^N \phi(Z_i, a)$, where $\phi(Z_i, a) = \frac{\exp\left(-\|Z_i - C_a\|_2^2\right)}{\sum_{t=1}^A \exp\left(-\|Z_i - C_t\|_2^2\right)}$. Similarly, we define the probability that the input node feature $X$ belongs to class $b$ as $\Pr[X \in b] = \frac{1}{n} \sum_{i=1}^n \phi(X_i, b)$, with $\phi(X_i, b) = \frac{\exp\left(-\left\|X_i - C_b'\right\|_2^2\right)}{\sum_{t=1}^B \exp\left(-\left\|X_i - C_t'\right\|_2^2\right)}$. More-over, the joint probabilities are calculated by $\Pr[Z \in a, X \in b] = \frac{1}{N} \sum_{i=1}^N \phi(Z_i, a)\phi(X_i, b)$ and $\Pr[Z \in a, Y = y] = \frac{1}{N} \sum_{i=1}^N \phi(Z_i, a)\mathbb{1}_{\{y_i = y\}}$, where $\mathbb{1}_{\{\}}$ is an indicator function. As a result, the mutual information $I(Z, X)$ and $I(Z, Y)$ can be computed by

$$I(Z, X) = \sum_{a=1}^A \sum_{b=1}^B \Pr[Z \in a, X \in b] \ln \frac{\Pr[Z \in a, X \in b]}{\Pr[Z \in a] \Pr[X \in b]}, \quad I(Z, Y) = \sum_{a=1}^A \sum_{y=1}^C \Pr[Z \in a, Y = y] \ln \frac{\Pr[Z \in a, Y = y]}{\Pr[Z \in a] \Pr[Y = y]},$$

and the IB loss $\text{IB}(Z, X, Y)$ can be computed by

$$\text{IB}(Z, X, Y) = \sum_{a=1}^A \sum_{b=1}^B \Pr[Z \in a, X \in b] \ln \frac{\Pr[Z \in a, X \in b]}{\Pr[Z \in a] \Pr[X \in b]} - \sum_{a=1}^A \sum_{y=1}^C \Pr[Z \in a, Y = y] \ln \frac{\Pr[Z \in a, Y = y]}{\Pr[Z \in a] \Pr[Y = y]}.$$

**Theorem 3.1.** Suppose $Z = BF$, with $B$ being the attention weight matrix and $F$ being the hidden node feature before applying the graph attention operation. For simplicity, we denote $\text{IB}(Z, X, Y)$ or $\text{IB}(BF, X, Y)$ by $\text{IB}(B)$. At step $\ell$ of gradient descent on $\text{IB}(B) = I(BF, X) - I(BF, Y)$, we have

$$B^{(\ell)} = B^{(\ell-1)} - \eta \nabla_B \text{IB}(B^{(\ell-1)}) = B^{(\ell-1)} - \eta Q^{(\ell-1)} \cdot F^\top. \tag{1}$$

where $\eta$ is the learning rate. $Q^{(\ell-1)} = \nabla_Z I(Z^{(\ell-1)}, X) - \nabla_Z I(Z^{(\ell-1)}, Y)$, where $Z^{(\ell-1)} = B^{(\ell-1)} F$. Formulas of $\nabla_Z I(Z, X)$ and $\nabla_Z I(Z, Y)$ are deferred to Lemma A.1 and Lemma A.2 in Section A of the appendix.

The proof of Theorem 3.1 is deferred to Section A of the appendix. Inspired by Theorem 3.1, we can understand a GNN with graph attention operations at multiple layers as an interactive process which reduces $\text{IB}(B)$ by gradient descent. The $\ell$-th graph attention layer simulates one step of gradient descent on $\text{IB}(B)$ according to Equation (1). Based on the gradient descent formulation, we design the formulation of the RGA-IB network, whose attention weight matrix $B^{(\ell)}$ at the $\ell$-th RGA-IB layer is generated from $B^{(\ell-1)}$, the attention weight matrix of the previous RGA-IB layer. Figure 2 illustrates the overall framework of an RGA-IB network and the structure of the RGA-IB layer. We present the training algorithm of the RGA-IB network in Algorithm 1, where $L$ denotes the number of RGA-IB layers.

## 4 EXPERIMENTS

In this section, we perform empirical evaluations of RGA-IB on public graph benchmarks Cora, Citeseer, Pubmed (Sen et al., 2008), and Polblogs (Adamic & Glance, 2005) for semi-supervised node classification under graph adversarial attacks. Implementation details of our experiments are presented in Section 4.1. Experiment results for semi-supervised node classification under adversarial attacks are presented in Section 4.2. Comprehensive ablation studies on IB loss at different layers of RGA-IB and the effects of RGA-IB in reducing IB loss are presented in Section 4.3 of the appendix. In addition, further implementation details on datasets, training settings, and attack

---

**Algorithm 1** Training Algorithm of the RGA-IB network

---

**Input:** The number of training epochs $t_{\text{train}}$, the number of warm-up epochs $t_{\text{warm}}$,
**Output:** The weights $\mathcal{W}$ of the RGA-IB network.
1: Initialize the weights of the RGA-IB network by $\mathcal{W} = \mathcal{W}(0)$.
2: Initialize the attention weight matrices $\left\{ B^{(\ell)} \right\}_{\ell=1}^{L}$ of all RGA-IB layers to identity matrices.
3: **for** $t \leftarrow 1$ to $t_{\text{train}}$ **do**
4:    **if** $t < t_{\text{warm}}$ **then**
5:       Perform gradient descent by a standard step of SGD on the cross-entropy loss to update the weights in the RGA-IB network with fixed attention weight matrices.
6:    **else**
7:       Update $\phi(Z_i, a)$ for all the clusters $a \in [C_a]$ and $i \in [N]$.
8:       **Forward step**: compute the attention weight matrices $\left\{ B^{(\ell)} \right\}_{\ell=1}^{L}$ for all the RGA-IB layers by Equation (1) using the updated $\left\{ \mathcal{C}_a^{(t)} \right\}_{a=1}^{C}$, and compute $\{Z_i\}_{i=1}^{N}$.
9:       **Backward step**: perform gradient descent by a standard step of SGD on the cross-entropy loss to update the weights in the RGA-IB network.
10:       Compute the class centroids $\{\mathcal{C}_a^{(t)}\}_{a=1}^{C}$ with the updated node representations $\{Z_i\}_{i=1}^{N}$.
11:    **end if**
12: **end for**
13: **return** The trained weights $\mathcal{W}$ of the RGA-IB network.

---

Table 1: Node classification performance (Accuracy±Std) under non-targeted attack Metattack (Zügner & Günnemann, 2019). The best result is highlighted in bold, and the second-best result is underlined. This convention is followed by all the tables in this paper. The results of RGA-IB are followed by the improvements over the best baselines.

| Dataset | Ptb Rate (%) | GCN | GAT | RGCN | UAG | Hang | GIB | URGL | RG-GIB | Differmer | GAR | CORNs | Pro-GNN | RGA-IB (Ours) |
|---|---|---|---|---|---|---|---|---|---|---|---|---|---|---|
| Cora | 0 | 83.5±0.4 | 83.9±0.6 | 83.0±0.4 | 82.0±0.5 | 80.0±0.3 | 82.2±0.6 | 82.1±0.6 | 82.1±0.6 | 84.9±0.6 | 83.2±0.6 | 82.5±0.4 | 82.9±0.2 | **85.0** (↑ 0.1) ±1.3 |
| | 5 | 75.5±0.4 | 77.0±0.7 | 75.0±1.3 | 76.2±1.2 | 76.9±1.2 | 75.8±1.2 | 75.3±1.2 | 76.8±1.2 | 77.0±1.3 | 75.6±1.1 | 75.9±1.8 | 77.6±1.9 | **79.2** (↑ 1.6) ±1.9 |
| | 10 | 72.0±1.3 | 74.1±1.8 | 73.1±1.3 | 76.0±1.8 | 76.8±1.5 | 74.0±1.8 | 73.0±1.8 | 75.9±1.8 | 74.0±1.6 | 73.7±1.3 | 74.6±2.0 | 77.3±1.9 | **79.1** (↑ 1.8) ±1.2 |
| | 15 | 69.4±1.7 | 70.6±2.7 | 71.6±1.3 | 74.9±1.3 | 75.6±1.6 | 72.9±1.5 | 72.4±1.5 | 74.3±1.5 | 72.8±1.3 | 71.8±1.7 | 72.5±1.2 | 75.4±1.3 | **76.9** (↑ 1.3) ±1.7 |
| | 20 | 67.6±1.0 | 68.8±2.3 | 67.9±2.2 | 71.9±2.2 | 72.2±1.3 | 70.0±1.8 | 69.6±1.8 | 72.0±1.8 | 70.6±2.2 | 71.9±2.4 | 71.9±1.5 | 71.3±1.8 | **74.2** (↑ 2.0) ±1.1 |
| | 25 | 64.8±1.1 | 65.5±2.5 | 66.0±2.4 | 68.0±2.4 | 69.1±1.3 | 67.9±2.2 | 67.2±2.2 | 68.9±2.2 | 66.3±2.1 | 68.7±2.8 | 69.4±2.9 | 68.7±1.5 | **71.0** (↑ 1.6) ±1.1 |
| Citeseer | 0 | 71.9±0.5 | 73.2±0.8 | 71.2±0.8 | 72.1±0.6 | 73.2±0.3 | 71.5±0.1 | 71.8±0.1 | 72.5±0.1 | 73.3±0.7 | 73.1±0.3 | 72.6±0.4 | 73.2±0.6 | **74.4** (↑ 1.1) ±1.0 |
| | 5 | 68.0±0.6 | 68.2±0.9 | 68.2±0.8 | 71.3±0.8 | 69.4±2.2 | 68.7±1.3 | 67.7±1.3 | 69.5±1.3 | 69.8±1.5 | 68.6±0.8 | 68.1±1.6 | 71.2±1.1 | **72.7** (↑ 1.5) ±2.0 |
| | 10 | 64.1±1.3 | 66.7±1.4 | 65.9±1.9 | 67.9±1.9 | 67.5±2.2 | 66.6±1.4 | 65.6±1.4 | 68.0±1.4 | 66.6±2.0 | 65.6±1.4 | 66.2±1.5 | 68.2±1.1 | **69.7** (↑ 1.5) ±1.7 |
| | 15 | 61.1±1.4 | 63.6±1.5 | 64.6±1.7 | 66.4±1.7 | 66.9±2.0 | 65.9±1.2 | 64.9±1.2 | 66.9±1.2 | 65.4±1.3 | 64.9±1.8 | 65.4±2.0 | 67.0±1.9 | **68.5** (↑ 1.6) ±1.7 |
| | 20 | 60.3±1.4 | 61.9±1.4 | 62.9±2.0 | 64.0±2.0 | 66.4±2.0 | 64.7±1.4 | 64.5±1.4 | 65.3±1.4 | 64.0±1.4 | 64.8±2.5 | 65.5±1.5 | 65.9±2.3 | **67.4** (↑ 1.0) ±1.9 |
| | 25 | 59.0±1.4 | 59.1±2.2 | 60.9±2.4 | 62.9±2.4 | 65.1±2.8 | 63.2±2.2 | 63.5±2.2 | 64.2±2.2 | 63.0±2.3 | 64.8±2.3 | 65.2±2.2 | 64.9±1.7 | **66.7** (↑ 1.5) ±1.9 |
| Polblogs | 0 | 95.6±0.3 | 95.3±0.2 | 95.2±0.1 | 90.1±2.2 | 94.7±1.0 | 95.4±0.8 | 95.5±0.8 | 95.2±0.8 | 95.7±0.2 | 95.5±0.1 | 95.3±0.8 | 93.2±0.6 | **96.5** (↑ 0.8) ±1.1 |
| | 5 | 87.7±0.4 | 88.4±0.2 | 89.2±0.1 | 90.3±0.1 | 90.3±1.3 | 90.2±0.2 | 90.5±0.2 | 90.0±0.2 | 90.0±2.2 | 90.5±0.2 | 89.3±0.3 | 90.9±1.4 | **92.0** (↑ 1.1) ±1.2 |
| | 10 | 84.6±1.7 | 85.9±1.4 | 85.9±1.9 | 87.0±1.9 | 86.7±1.3 | 86.0±1.3 | 85.8±1.3 | 86.6±1.3 | 85.6±1.7 | 86.3±1.4 | 87.3±1.7 | 87.1±1.4 | **89.1** (↑ 1.8) ±1.4 |
| | 15 | 71.6±1.7 | 72.0±1.1 | 72.1±1.8 | 81.9±1.8 | 82.3±1.6 | 79.5±1.9 | 79.2±1.9 | 82.1±1.9 | 79.9±1.8 | 83.4±1.7 | 82.0±1.4 | 83.3±1.9 | **85.0** (↑ 1.6) ±1.3 |
| | 20 | 65.0±1.0 | 67.1±1.2 | 67.3±1.2 | 71.5±1.2 | 72.6±2.1 | 69.4±1.3 | 69.2±1.3 | 69.4±1.3 | 71.3±1.8 | 69.8±1.2 | 69.9±1.2 | 72.5±1.7 | **73.8** (↑ 1.2) ±1.3 |
| | 25 | 64.0±2.4 | 64.2±2.1 | 66.1±2.1 | 69.0±2.1 | 70.8±2.3 | 67.2±1.4 | 65.2±1.4 | 68.2±1.4 | 68.4±2.0 | 67.5±1.9 | 66.5±2.7 | 69.4±2.0 | **71.9** (↑ 1.1) ±1.6 |
| Pubmed | 0 | 87.1±0.0 | 83.7±0.4 | 86.1±0.1 | 87.0±0.0 | 85.0±0.2 | 85.6±0.1 | 85.3±0.1 | 84.6±0.1 | 87.3±0.1 | 87.2±0.2 | 86.4±0.7 | 87.3±0.1 | **88.3** (↑ 1.0) ±1.2 |
| | 5 | 79.2±0.1 | 80.7±0.4 | 80.2±0.2 | 82.4±0.2 | 82.2±2.0 | 81.4±0.2 | 81.2±0.2 | 82.4±0.2 | 80.0±0.3 | 81.6±0.6 | 82.9±0.8 | 83.1±1.7 | **84.2** (↑ 1.1) ±1.9 |
| | 10 | 75.4±1.4 | 76.9±1.6 | 77.9±1.4 | 80.5±1.4 | 80.6±1.8 | 80.3±1.3 | 80.1±1.3 | 80.8±1.3 | 78.4±1.7 | 80.4±1.6 | 80.7±1.6 | 81.3±1.6 | **82.5** (↑ 1.2) ±1.7 |
| | 15 | 72.0±1.6 | 72.3±1.6 | 75.8±1.6 | 76.9±1.6 | 77.3±1.4 | 75.2±1.2 | 76.0±1.2 | 76.9±1.2 | 74.0±1.8 | 75.7±1.2 | 76.8±1.8 | 77.1±2.3 | **78.7** (↑ 1.4) ±1.2 |
| | 20 | 68.5±0.7 | 69.0±1.7 | 70.0±1.5 | 71.6±1.5 | 73.2±2.8 | 72.0±1.5 | 72.2±1.5 | 73.0±1.5 | 71.3±2.6 | 72.8±1.8 | 71.2±1.5 | 72.9±2.1 | **74.6** (↑ 1.4) ±1.0 |
| | 25 | 69.0±2.0 | 68.2±1.0 | 69.0±2.2 | 70.9±2.2 | 72.7±2.1 | 71.0±2.3 | 71.5±2.3 | 72.8±2.3 | 70.0±2.5 | 71.6±2.5 | 70.7±2.8 | 71.7±2.7 | **73.9** (↑ 1.2) ±1.3 |

settings are deferred to Section B of the appendix. Additional ablation study on the effectiveness of global node correlation learning in RGA-IB are deferred to Section C.1 of the appendix.

### 4.1 IMPLEMENTATION DETAILS

Following the settings in existing works on graph adversarial attacks (Jin et al., 2020; Zhao et al., 2023), we evaluate our method and competing baselines on the largest connected component (LCC) of the graph datasets. Details on the statistics of the datasets are deferred to Table 6 in Section B.1 of the supplementary. In our experiments, we randomly choose $10\%$ of nodes for training, $10\%$ of nodes for validation, and the remaining $80\%$ of nodes for testing following (Jin et al., 2020) on Cora, Citeseer, Polblogs, and Pubmed. For the training of the RGA-IB network, we first warm up the training of the network parameters for 100 epochs by only optimizing the weights of the linear layers and fixing the attention weight matrices as identity matrices. Following that, we train all the network weights in the RGA-IB network for 500 epochs. Adam is used as the optimizer for the training. Additional training settings and implementation details are deferred to Section B.2 of the appendix.

### 4.2 SEMI-SUPERVISED NODE CLASSIFICATION UNDER ADVERSARIAL ATTACKS

In our experiments for semi-supervised node classification under graph adversarial attacks, RGA-IB is compared with GCN (Kipf & Welling, 2017), GAT (Veličković et al., 2018), RGCN (Zhu et al.,

Table 2: Node classification performance (Accuracy±Std) under targeted attack Nettack (Zügner et al., 2018). The results of RGA-IB are followed by the improvements over the best baselines.

| Dataset | Attack Budget | GCN | GAT | RGCN | UAG | Hang | GIB | URGL | RG-GIB | Differmer | GAR | GCORNs | Pro-GNN | RGA-IB (Ours) |
|---|---|---|---|---|---|---|---|---|---|---|---|---|---|---|
| Cora | 0 | 81.4±1.0 | 82.1±1.1 | 81.3±1.6 | 82.3±1.1 | 80.8±1.3 | 81.9±2.8 | 81.3±1.8 | 83.1±2.2 | 84.9±1.8 | 83.3±0.7 | 82.6±0.4 | 84.8±0.6 | 86.1 (↑ 1.2) ±1.1 |
| | 1 | 75.1±1.0 | 76.0±2.1 | 76.8±1.7 | 81.8±1.2 | 77.0±3.2 | 80.2±2.8 | 80.0±1.5 | 80.6±2.3 | 80.3±1.6 | 81.3±1.2 | 81.0±1.1 | 83.8±0.5 | 85.5 (↑ 1.7) ±1.4 |
| | 2 | 70.6±1.1 | 70.2±1.4 | 71.0±1.1 | 78.0±1.8 | 76.5±2.6 | 77.2±2.2 | 75.9±2.2 | 77.9±1.3 | 78.5±1.7 | 75.9±1.3 | 78.4±1.3 | 78.7±0.7 | 80.7 (↑ 2.0) ±1.2 |
| | 3 | 68.0±1.7 | 65.5±1.3 | 66.5±1.6 | 72.5±1.1 | 73.1±2.9 | 72.9±3.4 | 71.9±1.4 | 72.9±1.1 | 73.1±1.5 | 73.5±1.1 | 73.1±0.7 | 72.4±0.5 | 74.7 (↑ 1.2) ±1.4 |
| | 4 | 61.6±1.5 | 61.7±0.9 | 59.3±2.7 | 70.3±2.5 | 72.5±2.1 | 69.9±2.1 | 70.9±2.1 | 70.5±1.2 | 70.5±1.2 | 70.9±1.2 | 71.2±0.6 | 70.1±0.7 | 73.6 (↑ 1.1) ±1.2 |
| | 5 | 55.5±1.7 | 58.3±2.0 | 55.3±1.7 | 66.3±1.2 | 68.8±2.6 | 66.4±2.4 | 64.4±2.4 | 67.9±2.4 | 67.4±1.5 | 67.2±1.2 | 66.8±1.0 | 66.9±0.7 | 69.9 (↑ 1.1) ±1.9 |
| Citeseer | 0 | 81.0±1.4 | 82.3±2.0 | 80.3±1.1 | 81.3±1.1 | 81.1±1.1 | 80.6±3.0 | 80.6±1.0 | 81.6±3.0 | 82.3±1.2 | 81.4±1.3 | 81.9±1.1 | 82.1±0.8 | 83.4 (↑ 1.1) ±1.7 |
| | 1 | 78.4±1.6 | 81.3±1.4 | 78.3±0.7 | 80.3±0.7 | 79.1±1.4 | 79.1±2.8 | 79.0±2.8 | 80.1±2.8 | 80.8±2.5 | 80.4±2.8 | 80.7±0.8 | 81.8±0.8 | 82.9 (↑ 1.1) ±1.9 |
| | 2 | 74.9±3.5 | 77.4±4.9 | 75.4±2.0 | 79.4±2.0 | 77.9±2.3 | 78.3±3.2 | 78.2±1.2 | 79.3±1.4 | 79.4±0.2 | 79.9±0.9 | 79.6±2.8 | 81.3±1.0 | 82.5 (↑ 1.2) ±1.4 |
| | 3 | 64.0±3.7 | 60.9±3.0 | 60.3±1.2 | 78.3±1.2 | 77.1±2.5 | 78.7±3.3 | 78.1±1.3 | 78.7±1.5 | 77.1±2.2 | 77.9±2.6 | 78.1±2.0 | 79.7±2.0 | 81.0 (↑ 1.3) ±1.6 |
| | 4 | 55.4±2.6 | 61.6±4.6 | 55.5±1.8 | 77.5±1.8 | 78.4±1.6 | 77.7±6.5 | 77.4±2.5 | 78.1±2.1 | 69.2±3.4 | 69.4±3.9 | 77.7±1.6 | 77.8±2.8 | 79.5 (↑ 1.1) ±1.6 |
| | 5 | 47.6±5.2 | 55.6±6.3 | 47.4±2.0 | 71.4±2.0 | 73.5±3.5 | 71.0±4.6 | 70.5±2.5 | 72.0±1.6 | 69.2±2.3 | 69.7±2.6 | 71.6±2.7 | 71.3±5.0 | 74.6 (↑ 1.1) ±1.9 |
| Polblogs | 0 | 97.0±0.2 | 97.3±0.3 | 97.0±0.1 | 97.1±0.1 | 97.4±0.5 | 97.2±0.8 | 97.3±0.3 | 97.3±0.8 | 97.2±0.4 | 97.4±0.8 | 97.5±0.2 | 97.1±0.2 | 98.2 (↑ 0.7) ±1.6 |
| | 1 | 96.8±0.2 | 97.2±0.3 | 97.0±0.1 | 96.0±0.1 | 97.4±0.4 | 97.1±0.3 | 97.1±0.3 | 97.2±0.5 | 97.4±0.2 | 97.5±0.4 | 97.6±0.5 | 96.8±0.1 | 98.1 (↑ 0.5) ±1.9 |
| | 2 | 95.6±0.2 | 96.1±0.7 | 95.9±0.2 | 97.0±0.2 | 96.9±0.2 | 95.3±0.7 | 95.3±0.5 | 95.9±0.3 | 97.1±0.2 | 95.9±0.3 | 96.0±0.4 | 97.2±0.1 | 97.8 (↑ 0.6) ±1.5 |
| | 3 | 95.4±0.2 | 95.8±0.6 | 95.6±0.3 | 96.6±0.3 | 96.7±0.2 | 95.4±0.1 | 95.4±0.1 | 95.9±0.1 | 96.9±0.9 | 95.3±0.7 | 95.6±1.0 | 96.9±0.1 | 97.5 (↑ 0.6) ±1.1 |
| | 4 | 94.2±0.2 | 94.8±0.7 | 94.4±0.3 | 96.1±0.3 | 96.3±0.5 | 94.9±0.6 | 94.9±0.6 | 95.3±0.5 | 96.8±0.6 | 94.1±0.2 | 94.4±0.5 | 96.9±0.2 | 97.4 (↑ 0.5) ±1.3 |
| | 5 | 93.0±0.5 | 93.3±1.4 | 93.2±0.4 | 95.6±0.4 | 95.9±0.3 | 93.2±0.4 | 93.2±0.4 | 94.6±0.4 | 95.1±0.8 | 93.6±0.7 | 93.4±0.8 | 96.1±0.3 | 97.2 (↑ 1.1) ±1.5 |
| Pubmed | 0 | 88.1±1.4 | 87.0±1.1 | 84.9±1.4 | 87.3±1.4 | 85.4±1.2 | 85.9±1.1 | 85.4±1.1 | 86.8±1.1 | 87.4±1.8 | 84.6±1.5 | 84.5±1.3 | 88.5±1.2 | 89.6 (↑ 1.1) ±1.9 |
| | 1 | 87.0±1.6 | 88.4±2.2 | 83.8±1.7 | 86.6±1.7 | 84.4±2.0 | 84.6±2.9 | 84.5±1.6 | 85.5±2.9 | 86.0±2.3 | 83.8±1.6 | 83.8±1.9 | 87.6±2.0 | 88.8 (↑ 1.2) ±1.4 |
| | 2 | 84.1±3.5 | 83.9±1.4 | 84.2±1.1 | 85.2±1.1 | 84.4±3.1 | 84.3±3.5 | 84.1±3.5 | 85.3±2.5 | 86.0±2.2 | 84.3±1.5 | 84.4±1.7 | 85.9±3.1 | 87.0 (↑ 1.0) ±1.3 |
| | 3 | 81.3±3.7 | 81.3±1.3 | 82.5±1.6 | 83.9±1.6 | 84.0±3.1 | 84.9±3.3 | 84.3±3.3 | 84.2±1.3 | 81.8±3.5 | 82.3±1.1 | 82.2±1.4 | 84.4±3.1 | 85.6 (↑ 1.2) ±1.9 |
| | 4 | 76.4±2.6 | 78.5±0.9 | 80.0±2.7 | 80.0±2.3 | 79.7±3.3 | 79.4±3.2 | 78.0±3.2 | 79.9±1.2 | 77.4±2.4 | 80.0±2.1 | 81.0±2.7 | 80.0±3.3 | 81.6 (↑ 1.6) ±1.2 |
| | 5 | 68.3±5.2 | 74.3±2.0 | 75.1±3.7 | 75.2±1.4 | 70.6±4.3 | 70.2±1.6 | 71.2±1.6 | 70.6±1.6 | 71.3±2.2 | 75.1±3.4 | 75.2±1.1 | 72.2±4.3 | 76.8 (↑ 1.6) ±1.3 |

2019), UAG (Feng et al., 2021), HANG (Zhao et al., 2023), Pro-GNN (Jin et al., 2020), GIB (Wu et al., 2020), UGRL (Wang et al., 2023b), RG-GIB (Dai et al., 2023a), Difformer (Wu et al., 2023a), GAR (Fountoulakis et al., 2023), and GCORNs (Abbahaddou et al., 2024). Among all the compared methods, GAT, RGCN, UAG, Difformer, and GAR are attention-based GNNs, and RGCN, UAG, and GAR are specifically designed for semi-supervised node classification under graph adversarial attacks. GIB, UGRL, and RG-GIB are robust learning methods designed by the IB principle.

In this section, we summarize the experiment results of our proposed RGA-IB by comparing the semi-supervised node classification accuracy between the baseline methods and our proposed RGA-IB under different types of graph adversarial attacks with different attack strengths. Detaisl on the attack settings are deferred to Section B.3 of the appendix. The results for Metattack (Zügner & Günnemann, 2019), Nettack (Zügner et al., 2018), and Topology Attack (Xu et al., 2019a) are shown in Table 1, Table 2, and Table 3, respectively. We run all experiments ten times and report the mean and standard deviation of the node classification accuracy. It is observed from the results that RGA-IB significantly outperforms existing robust graph learning methods under different adversarial attacks. For instance, the average improvements of RGA-IB over the second-best methods across different attack budgets on Pubmed for Metattack, Nettack, and Topology Attack are $1.46\%$, $1.54\%$, and $1.48\%$, demonstrating that RGA-IB successfully reduces the negative effects of both noisy edges and nodes by reducing the IB loss with the robust graph attention design.

Table 3: Node classification performance (Accuracy±Std) under Topology Attack (Xu et al., 2019a). The results of RGA-IB are followed by the improvements over the best baselines.

| Dataset | Ptb Rate (%) | GCN | GAT | RGCN | UAG | Hang | GIB | URGL | RG-GIB | Difformer | GAR | CORNs | Pro-GNN | RGA-IB (Ours) |
|---|---|---|---|---|---|---|---|---|---|---|---|---|---|---|
| Cora | 0 | 83.5±0.4 | 84.0±0.7 | 83.1±0.4 | 82.1±0.5 | 80.1±0.3 | 82.2±0.7 | 82.1±0.7 | 83.2±0.7 | 84.9±0.7 | 83.3±0.7 | 82.6±0.4 | 83.0±0.2 | 85.0 (↑ 0.1) ±1.3 |
| | 5 | 75.5±0.4 | 77.0±0.7 | 75.0±1.3 | 76.2±1.2 | 76.9±1.2 | 75.8±1.2 | 75.3±1.2 | 76.8±1.2 | 77.0±1.3 | 75.6±1.1 | 75.9±1.8 | 77.6±1.9 | 79.6 (↑ 2.0) ±1.9 |
| | 10 | 72.0±1.3 | 74.1±1.8 | 73.1±1.3 | 76.0±1.8 | 76.8±1.5 | 74.0±1.8 | 73.0±1.8 | 75.9±1.8 | 74.0±1.6 | 73.7±1.3 | 74.6±2.0 | 77.3±1.9 | 79.3 (↑ 2.0) ±1.2 |
| | 15 | 69.4±1.7 | 70.6±2.7 | 71.6±1.3 | 74.9±1.3 | 75.6±1.6 | 72.9±1.5 | 72.4±1.5 | 74.3±1.5 | 72.8±1.3 | 71.8±1.7 | 72.5±1.2 | 75.4±1.3 | 76.9 (↑ 1.3) ±1.7 |
| | 20 | 67.6±1.0 | 68.8±2.3 | 67.9±2.2 | 71.9±2.2 | 72.2±1.3 | 70.0±1.8 | 69.6±1.8 | 72.0±1.8 | 70.6±2.2 | 71.9±2.4 | 71.9±1.5 | 71.3±1.8 | 74.5 (↑ 2.3) ±1.1 |
| | 25 | 64.8±1.1 | 65.5±2.5 | 66.0±2.4 | 68.0±2.4 | 69.1±1.3 | 67.9±2.2 | 67.2±2.2 | 68.9±2.2 | 66.3±2.1 | 68.7±2.8 | 69.4±2.9 | 68.7±1.5 | 71.0 (↑ 1.6) ±1.1 |
| Citeseer | 0 | 72.0±0.6 | 73.3±0.8 | 71.2±0.8 | 72.1±0.6 | 73.3±0.4 | 71.5±0.2 | 71.4±0.2 | 72.5±0.2 | 73.3±0.8 | 73.1±0.3 | 72.7±0.5 | 73.3±0.7 | 74.5 (↑ 1.2) ±1.0 |
| | 5 | 68.0±0.6 | 68.2±0.9 | 68.2±0.8 | 71.3±0.8 | 69.4±2.2 | 68.7±1.3 | 67.7±1.3 | 69.5±1.3 | 69.8±1.5 | 68.6±0.8 | 68.1±1.6 | 71.2±1.1 | 72.7 (↑ 1.5) ±2.0 |
| | 10 | 64.1±1.3 | 66.7±1.4 | 65.9±1.9 | 67.9±1.9 | 67.5±2.2 | 66.6±1.4 | 65.6±1.4 | 68.0±1.4 | 66.6±2.0 | 65.6±1.4 | 66.2±1.5 | 68.2±1.1 | 69.7 (↑ 1.5) ±1.7 |
| | 15 | 61.1±1.4 | 63.6±1.5 | 64.6±1.7 | 66.4±1.7 | 66.9±2.0 | 65.9±1.2 | 64.9±1.2 | 66.9±1.2 | 65.4±1.3 | 64.9±1.8 | 65.4±2.0 | 67.0±1.9 | 68.5 (↑ 1.5) ±1.7 |
| | 20 | 60.3±1.4 | 61.9±1.4 | 62.9±2.0 | 64.0±2.0 | 66.4±2.0 | 64.7±1.4 | 64.5±1.4 | 65.3±1.4 | 64.0±1.4 | 64.8±2.5 | 65.5±1.5 | 65.9±2.3 | 67.4 (↑ 1.0) ±1.9 |
| | 25 | 59.0±1.4 | 59.1±2.2 | 60.9±2.4 | 62.9±2.4 | 65.1±2.8 | 63.4±2.3 | 63.5±2.2 | 64.2±2.2 | 63.8±2.3 | 64.3±2.3 | 65.2±2.2 | 64.9±1.7 | 66.7 (↑ 1.5) ±1.9 |
| Polblogs | 0 | 95.7±0.4 | 95.4±0.2 | 95.2±0.1 | 90.1±2.2 | 94.8±1.1 | 95.4±0.8 | 95.3±0.8 | 95.1±0.8 | 95.7±0.2 | 95.6±0.2 | 95.3±0.8 | 93.2±0.6 | 96.5 (↑ 0.8) ±1.1 |
| | 5 | 87.7±0.4 | 88.4±0.2 | 89.2±0.1 | 90.3±0.1 | 90.3±1.3 | 90.2±0.2 | 90.5±0.2 | 90.0±0.2 | 90.0±2.2 | 90.5±0.2 | 89.3±0.3 | 90.9±1.4 | 92.0 (↑ 1.1) ±1.2 |
| | 10 | 84.6±1.7 | 85.9±1.4 | 85.9±1.9 | 87.0±1.9 | 86.7±1.3 | 86.0±1.3 | 85.8±1.3 | 86.6±1.3 | 85.6±1.7 | 86.3±1.4 | 87.3±1.7 | 87.1±1.4 | 89.1 (↑ 1.8) ±1.4 |
| | 15 | 71.6±1.7 | 72.0±1.1 | 72.1±1.8 | 81.9±1.8 | 82.3±1.6 | 79.5±1.7 | 79.2±1.9 | 82.1±1.9 | 79.9±1.8 | 83.4±1.7 | 82.0±1.4 | 83.3±1.9 | 85.0 (↑ 1.6) ±1.3 |
| | 20 | 65.0±1.0 | 67.1±1.2 | 67.3±1.2 | 71.5±1.2 | 72.6±2.1 | 69.4±1.3 | 69.2±1.3 | 69.4±1.3 | 71.3±1.8 | 69.8±1.2 | 69.9±1.2 | 72.5±1.7 | 73.8 (↑ 1.2) ±1.3 |
| | 25 | 64.0±2.4 | 64.2±2.1 | 66.1±2.1 | 69.0±2.1 | 70.8±2.3 | 67.2±1.4 | 65.2±1.4 | 68.2±1.4 | 68.4±2.0 | 67.5±1.9 | 66.5±2.7 | 69.4±2.0 | 71.9 (↑ 1.1) ±1.6 |
| Pubmed | 0 | 87.2±0.1 | 83.7±0.4 | 86.2±0.2 | 87.1±0.1 | 85.1±0.2 | 85.7±0.1 | 85.4±0.1 | 86.7±0.1 | 87.3±0.2 | 87.3±0.2 | 86.4±0.7 | 87.3±0.2 | 88.4 (↑ 1.1) ±1.3 |
| | 5 | 79.2±0.1 | 80.7±0.4 | 80.2±0.2 | 82.4±0.2 | 82.2±2.0 | 81.4±0.2 | 81.2±0.2 | 82.4±0.2 | 80.0±0.3 | 81.6±0.6 | 82.9±0.8 | 83.1±1.7 | 84.2 (↑ 1.1) ±1.9 |
| | 10 | 75.4±1.4 | 76.9±1.6 | 77.9±1.4 | 80.5±1.4 | 80.6±1.8 | 80.3±1.3 | 80.1±1.3 | 80.8±1.3 | 78.4±1.7 | 80.4±1.6 | 80.7±1.6 | 81.3±1.6 | 82.5 (↑ 1.2) ±1.7 |
| | 15 | 72.0±1.6 | 72.3±1.6 | 75.8±1.6 | 76.9±1.6 | 77.3±1.4 | 75.2±1.2 | 76.0±1.2 | 76.9±1.2 | 74.0±1.8 | 75.7±1.2 | 76.8±1.8 | 77.1±2.3 | 78.7 (↑ 1.4) ±1.2 |
| | 20 | 68.5±0.7 | 69.0±1.7 | 70.0±1.5 | 71.6±1.5 | 73.2±2.8 | 72.0±1.5 | 72.2±1.5 | 73.0±1.5 | 71.3±2.6 | 72.8±1.8 | 71.2±1.5 | 72.9±2.1 | 74.6 (↑ 1.4) ±1.0 |
| | 25 | 69.0±2.0 | 68.2±1.0 | 69.0±2.2 | 70.9±2.2 | 72.7±2.1 | 71.0±2.3 | 71.5±2.3 | 72.8±2.3 | 70.0±2.5 | 71.6±2.5 | 70.7±2.8 | 71.7±2.7 | 73.9 (↑ 1.2) ±1.3 |

## 4.3 ABLATION STUDY

**Study on the IB Loss at Different Layers of RGA-IB.** To study how the IB loss $IB(B)$ decreases with respect to layer index $\ell$ of an RGA-IB network, we calculate $IB(B)$ across different RGA-IB layers in the RGA-IB network. The study is performed on Cora and Citeseer under Metattack with a perturbation rate of $25\%$ with both a 2-layer RGA-IB network and a 4-layer RGA-IB network. We also calculate the IB loss at different layers of 2-layer Difformer, 4-layer Difformer, 2-layer GAR, and 4-layer GAR. It is observed from the results in Table 4 that the IB loss decreases in deeper layers with a larger layer index for both the 2-layer RGA-IB network and the 4-layer RGA-

IB network. This observation suggests that node features in deeper layers of RGA-IB networks correlate more closely with the class labels and less with the input node attributes, adhering to the IB principle. Moreover, the RGA-IB networks reduce the IB loss to lower levels in deeper layers compared to Difformer and GAR, demonstrating the superiority of RGA-IB over the existing state-of-the-art graph attention method and robust graph attention method. In addition, we observe that the 2-layer RGA-IB network already decreases the IB loss of node features to the same level as the 4-layer RGA-IB network, leading to similar node classification performance. Therefore we use 2-layer RGA-IB networks for all our experiments in this work as the 2-layer RGA-IB network costs less computational resources while enjoying the same level of effectiveness in reducing IB loss.

Table 4: Ablation study comparing the IB loss at different layers for RGA-IB. The study is performed on Cora and Citeseer under Metattack with a perturbation rate of $25\%$.

| Methods | Layer Number | Cora | | | | | Citeseer | | | | |
|---|---|---|---|---|---|---|---|---|---|---|---|
| | | IB Loss | | | | ACC | IB Loss | | | | ACC |
| | | Layer 1 | Layer 2 | Layer 3 | Layer 4 | | Layer 1 | Layer 2 | Layer 3 | Layer 4 | |
| Difformer | 2 | -0.096 | -0.115 | - | - | 61.68 | -0.088 | -0.118 | - | - | 63.57 |
| Difformer | 4 | -0.077 | -0.095 | -0.099 | -0.122 | 62.32 | -0.070 | -0.082 | -0.103 | -0.117 | 64.33 |
| GAR | 2 | -0.089 | -0.138 | - | - | 65.50 | -0.074 | -0.130 | - | - | 67.22 |
| GAR | 4 | -0.065 | -0.092 | -0.116 | -0.132 | 64.96 | -0.070 | -0.112 | -0.125 | -0.125 | 66.40 |
| RGA-IB | 2 | -0.134 | -0.214 | - | - | **71.43** | -0.096 | -0.189 | - | - | **70.94** |
| RGA-IB | 4 | -0.125 | -0.179 | -0.195 | -0.208 | 71.38 | -0.082 | -0.146 | -0.175 | -0.190 | 70.92 |

**Study on the Effects of RGA-IB in Reducing the IB Loss.** In this section, we evaluate the effectiveness of RGA-IB in minimizing the IB loss. We calculate the IB loss for both RGA-IB and baseline graph attention methods, including GAT (Veličković et al., 2018), UAG (Feng et al., 2021), RGCN (Zhu et al., 2019), Difformer (Wu et al., 2023a), and GAR (Fountoulakis et al., 2023) on Cora and Citeseer under Metattack with different perturbation rates. The results are shown in Table 5. It is observed from the results that the IB loss of graph attention methods correlates closely with the node classification accuracy under adversarial attacks, and the two methods with the lowest two IB losses always enjoy the top two best robust accuracies. Graph attention methods with lower IB loss better adhere to the IB principle. Moreover, the IB loss can be further decreased to a considerable extent by optimizing the IB loss explicitly in RGA-IB.

Table 5: Ablation study on the effects of RGA-IB in reducing the IB loss compared to the existing graph attention methods. Node classification accuracy for all methods on all the datasets is attached in parentheses after the IB loss.

| Dataset | Attack Budget | GAT | RGCN | UAG | Difformer | GAR | RGA-IB (Ours) |
|---|---|---|---|---|---|---|---|
| Cora | 0 | -0.105 (83.90) | -0.084 (83.04) | -0.103 (82.03) | -0.114 (84.90) | -0.096 (83.23) | **-0.127 (85.03)** |
| | 5 | -0.110 (80.40) | -0.094 (77.42) | -0.105 (79.13) | -0.109 (79.42) | -0.117 (80.22) | **-0.128 (83.82)** |
| | 10 | -0.102 (75.61) | -0.101 (72.22) | -0.110 (75.10) | -0.118 (77.55) | -0.122 (77.94) | **-0.144 (80.14)** |
| | 15 | -0.105 (69.73) | -0.102 (66.82) | -0.105 (71.03) | -0.117 (73.46) | -0.125 (75.14) | **-0.179 (77.42)** |
| | 20 | -0.109 (59.94) | -0.105 (59.17) | -0.110 (65.71) | -0.115 (65.97) | -0.133 (68.72) | **-0.186 (74.59)** |
| | 25 | -0.114 (54.70) | -0.100 (50.51) | -0.119 (60.82) | -0.122 (62.35) | -0.138 (65.50) | **-0.214 (71.43)** |
| Citeseer | 0 | -0.110 (73.23) | -0.080 (71.20) | -0.093 (72.10) | -0.124 (73.34) | -0.115 (73.13) | **-0.133 (74.43)** |
| | 5 | -0.102 (72.79) | -0.082 (70.50) | -0.090 (70.51) | -0.114 (70.40) | -0.119 (73.04) | **-0.141 (74.42)** |
| | 10 | -0.095 (70.63) | -0.084 (67.71) | -0.092 (69.54) | -0.088 (69.33) | -0.130 (71.53) | **-0.153 (73.60)** |
| | 15 | -0.115 (69.02) | -0.110 (65.59) | -0.100 (65.93) | -0.091 (68.70) | -0.119 (70.70) | **-0.172 (73.22)** |
| | 20 | -0.104 (61.04) | -0.108 (62.39) | -0.098 (59.30) | -0.112 (67.67) | -0.145 (69.43) | **-0.176 (72.21)** |
| | 25 | -0.102 (61.83) | -0.092 (55.33) | -0.110 (59.22) | -0.117 (64.32) | -0.130 (67.22) | **-0.189 (70.94)** |

## 5 CONCLUSION

In this work, we find that the IB loss of attention-based GNNs is a strong indicator of their robustness against graph adversarial attacks, and attention-based GNNs with lower IB loss learn node representations that correlate less with the input training data while aligning better with the target outputs. Due to better adhering to the IB principle, attention-based GNNs with lower IB loss usually show stronger robustness against graph adversarial attacks. Inspired by such observation, we propose a novel graph attention method termed Robust Graph Attention inspired by Information Bottleneck, or RGA-IB, which explicitly minimizes the IB loss of a multi-layer GNN through a carefully designed graph attention mechanism. Extensive experiment results show that RGA-IB networks exhibit lower IB loss and show significantly improved node classification accuracy under variant graph adversarial attacks compared to existing robust GNNs and robust attention-based GNNs.

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

## A   PROOF OF THEOREM 3.1

We need the following two lemmas before the proof of Theorem 3.1. It is noted $Z = BF$, with $B$ being the attention weight matrix and $F$ being the hidden node feature before applying the graph attention operation. We abbreviate $\text{IB}(Z, X, Y)$ as $\text{IB}(B)$.

**Lemma A.1.**

$$\nabla_B I(Z, X) = \nabla_Z I(Z, X) \cdot F^\top. \tag{2}$$

**Lemma A.2.**

$$\nabla_B I(Z, Y) = \nabla_Z I(Z, Y) \cdot F^\top. \tag{3}$$

***Proof of Theorem 3.1.*** We note that $IB(B) = I(Z, X) - I(Z, Y)$. Then $\nabla_B \text{IB}(B) = \nabla_B I(Z, X) - \nabla_B I(Z, Y)$. With Lemma A.1 and Lemma A.2, we have

$$\begin{aligned}
\nabla_B \text{IB}(B) &= \nabla_B I(BF, X) - \nabla_B I(BF, Y) \\
&= \nabla_Z I(Z, X) \cdot F^\top - \nabla_Z I(Z, Y) \cdot F^\top \\
&= (\nabla_Z I(Z, X) - \nabla_Z I(Z, Y)) \cdot F^\top.
\end{aligned}$$

Let $Q^{(\ell-1)} = \nabla_Z I(Z^{(\ell-1)}, X) - \nabla_Z I(Z^{(\ell-1)}, Y)$, we get $\nabla_B \text{IB}(B^{(\ell-1)}) = Q^{(\ell-1)} \cdot F^\top$. Therefore, at step $\ell$ of gradient descent on $\text{IB}(B)$, we have

$$B^{(\ell)} = B^{(\ell-1)} - \eta Q^{(\ell-1)} \cdot F^\top.$$

$\square$

***Proof of Lemma A.1.***

$$I(Z, X) = \frac{1}{n} \sum_{a=1}^{A} \sum_{b=1}^{B} \left( \sum_{i=1}^{n} \phi(Z_i, a)\phi(X_i, b) \right) \left( \ln n + \ln \sum_{j=1}^{n} \phi(Z_j, a)\phi(X_j, b) - \ln \sum_{k=1}^{n} \phi(Z_k, a) - \ln \sum_{m=1}^{n} \phi(X_m, b) \right)$$

Let $G_{ia} = \phi(Z_i, a)$, and $Z_{ib} = \phi(X_i, b)$, then

$$I(Z,X) = \frac{1}{n} \sum_{a=1}^{A} \sum_{b=1}^{B} \left( \sum_{i=1}^{n} G_{ia} Z_{ib} \right) \left( \ln n + \ln \sum_{j=1}^{n} G_{ja} Z_{jb} - \ln \sum_{k=1}^{n} G_{ka} - \ln \sum_{m=1}^{n} Z_{mb} \right)$$

Next, we have

$$\nabla_{Z_i} I(Z,X) = \frac{1}{n} \sum_{a=1}^{A} \sum_{b=1}^{B} \left( Z_{ib} U_{ia} \left( \ln n + \ln \sum_{j=1}^{n} G_{ja} Z_{jb} - \ln \sum_{k=1}^{n} G_{ka} - \ln \sum_{m=1}^{n} Z_{mb} \right) + \left( \sum_{i=1}^{n} G_{ia} Z_{ib} \right) \left( \frac{Z_{ib} U_{ia}}{\sum_{j=1}^{n} G_{ja} Z_{jb}} - \frac{U_{ia}}{\sum_{k=1}^{n} G_{ka}} \right) \right)$$

$$= \frac{1}{n} \sum_{a=1}^{A} \sum_{b=1}^{B} \left( Z_{ib} U_{ia} \left( \ln n + \ln \sum_{j=1}^{n} G_{ja} Z_{jb} - \ln \sum_{k=1}^{n} G_{ka} - \ln \sum_{m=1}^{n} Z_{mb} \right) + U_{ia} \left( Z_{ib} - \frac{\sum_{i=1}^{n} G_{ia} Z_{ib}}{\sum_{k=1}^{n} G_{ka}} \right) \right).$$

$$(4)$$

where $\nabla_{Z_i} I(Z,X)$ is the $i$-th row of $\nabla_Z I(Z,X)$. Define $G_{ia}$ as $\phi(Z_i, a)$, $Z_{ib}$ as $\phi(X_i, b)$, and $U_{ia}$ as $\nabla_{Z_i} G_{ia}$. Let $S_{ia} = \exp\left(-\|Z_i - C_a\|^2\right)$, $U_{ia}$ can be computed by

$$U_{ia} = \frac{-2(Z_i - C_a) S_{ia} \sum_{b=1}^{A} S_{ib} - S_{ia} \sum_{b=1}^{A} -2(Z_i - C_b) S_{ib}}{\left( \sum_{b=1}^{A} S_{ib} \right)^2} \qquad (5)$$

$$= -2 \left( \frac{S_{ia}(Z_i - C_a)}{\sum_{b=1}^{A} S_{ib}} - \frac{S_{ia}}{\sum_{b=1}^{A} S_{ib}} \sum_{b=1}^{A} \frac{S_{ib}(Z_i - C_b)}{\sum_{c=1}^{A} S_{ic}} \right)$$

$$= -2 \left( G_{ia}(Z_i - C_a) - G_{ia} \sum_{b=1}^{A} G_{ib}(Z_i - C_b) \right)$$

$$= -2 G_{ia} \left( Z_i - C_a - \sum_{b=1}^{A} G_{ib}(Z_i - C_b) \right)$$

Taking the value of $U_{ia}$ back to Equation (4), we can get $\nabla_{Z_i} I(Z,X)$. Note that $\nabla_{Z_i} I(Z,X) \in \mathbb{R}^{1 \times d}$. $\nabla_Z I(Z,X) \in \mathbb{R}^{n \times d}$, where the $i$-th row of $\nabla_Z I(Z,X)$ is $\nabla_{Z_i} I(Z,X)$. Given that $Z = BF$, we have

$$\nabla_B I(Z,X) = \nabla_Z I(Z,X) \cdot F^\top. \qquad (6)$$

$\square$

***Proof of Lemma A.2.***

$$I(Z,Y) = \frac{1}{n} \sum_{a=1}^{A} \sum_{c=1}^{C} \left( \sum_{i=1}^{n} \phi(Z_i, a) \mathbb{1}_{\{y_i = b\}} \right) \left( \ln n + \ln \sum_{j=1}^{n} \phi(Z_j, a) \mathbb{1}_{\{y_j = b\}} - \ln \sum_{k=1}^{n} \phi(Z_k, a) - \ln \sum_{m=1}^{n} \mathbb{1}_{\{y_m = c\}} \right)$$

By replacing the value of $Z_{ib}$ with $Z_{ib} = \mathbb{1}_{\{y_i = b\}}$, we can get the value of $\nabla_B I(Z,Y)$ following the formulation of $\nabla_B I(Z,X)$.

Let $G_{ia} := \phi(Z_i, a)$, then

$$I(Z,Y) = \frac{1}{n} \sum_{a=1}^{A} \sum_{b=1}^{B} \left( \sum_{i=1}^{n} G_{ia} \mathbb{1}_{\{y_i = b\}} \right) \left( \ln n + \ln \sum_{j=1}^{n} G_{ja} \mathbb{1}_{\{y_j = b\}} - \ln \sum_{k=1}^{n} G_{ka} - \ln \sum_{m=1}^{n} \mathbb{1}_{\{y_m = b\}} \right)$$

Next, we have

$$
\begin{aligned}
\nabla_{Z_i} I(Z,Y) &= \frac{1}{n} \sum_{a=1}^{A} \sum_{b=1}^{B} \left( \mathbb{1}_{\{y_i=b\}} U_{ia} \left( \ln n + \ln \sum_{j=1}^{n} G_{ja} \mathbb{1}_{\{y_j=b\}} - \ln \sum_{k=1}^{n} G_{ka} - \ln \sum_{m=1}^{n} \mathbb{1}_{\{y_m=b\}} \right) \right) \\
&+ \frac{1}{n} \sum_{a=1}^{A} \sum_{b=1}^{B} \left( \left( \sum_{i=1}^{n} G_{ia} Z_{ib} \right) \left( \frac{\mathbb{1}_{\{y_i=b\}} U_{ia}}{\sum_{j=1}^{n} G_{ja} Z_{jb}} - \frac{U_{ia}}{\sum_{k=1}^{n} G_{ka}} \right) \right) \\
&= \frac{1}{n} \sum_{a=1}^{A} \sum_{b=1}^{B} \left( \mathbb{1}_{\{y_i=b\}} U_{ia} \left( \ln n + \ln \sum_{j=1}^{n} G_{ja} \mathbb{1}_{\{y_j=b\}} - \ln \sum_{k=1}^{n} G_{ka} - \ln \sum_{m=1}^{n} \mathbb{1}_{\{y_m=b\}} \right) \right) \\
&+ \frac{1}{n} \sum_{a=1}^{A} \sum_{b=1}^{B} \left( U_{ia} \left( \mathbb{1}_{\{y_i=b\}} - \frac{\sum_{i=1}^{n} G_{ia} \mathbb{1}_{\{y_i=b\}}}{\sum_{k=1}^{n} G_{ka}} \right) \right),
\end{aligned}
\tag{7}
$$

where $U_{ia} = \nabla_{Z_i} G_{ia} = \nabla_{Z_i} \phi(Z_i, a) = -2 G_{ia} \left( Z_i - C_a - \sum_{b=1}^{A} G_{ib}(Z_i - C_b) \right)$. Taking the value of $U_{ia}$ back to Equation (7), we can get $\nabla_{Z_i} I(Z,Y)$. Note that $\nabla_{Z_i} I(Z,Y) \in \mathbb{R}^{1 \times d}$. $\nabla_Z I(Z,Y) \in \mathbb{R}^{n \times d}$, where the $i$-th row of $\nabla_Z I(Z,Y)$ is $\nabla_{Z_i} I(Z,Y)$. Given that $Z = BF$, we have

$$
\nabla_B I(Z,Y) = \nabla_Z I(Z,Y) \cdot F^\top.
\tag{8}
$$

$\square$

## B  MORE EXPERIMENT SETTINGS

### B.1  DATASETS

Following previous works on adversarial attacks and defense of GNNs (Jin et al., 2020; Zügner & Günnemann, 2019; Entezari et al., 2020), we evaluate RGA-IB on four public benchmark datasets for node classification, including three citation graphs, which are Cora, Citeseer, and Pubmed, and one blog graph, that is, Polblogs. Following previous works on graph adversarial attacks, we evaluate our method and baselines on the largest connected component (LCC) of the graphs. We show the statistics of the datasets in Table 6.

Table 6: Statistics of Cora, Citeseer, Polblogs, and Pubmed.

|          | # Node | # Edge | Classes | Features |
|----------|--------|--------|---------|----------|
| Cora     | 2,485  | 5,069  | 7       | 1,433    |
| Citeseer | 2,110  | 3,668  | 6       | 3,703    |
| Polblogs | 1,222  | 16,714 | 2       | 1,222    |
| Pubmed   | 19,717 | 44,338 | 3       | 500      |

### B.2  ADDITIONAL IMPLEMENTATION DETAILS

In our experiments for semi-supervised node classification, we search for the optimal values of different hyper-parameters, including learning rate, weight decay, hidden dimension, and dropout rate, by 5-fold cross-validation on the training data of each dataset. We search for the learning rate from $\{1 \times 10^{-4}, 5 \times 10^{-4}, 1 \times 10^{-3}, 5 \times 10^{-3}, 1 \times 10^{-2}, 3 \times 10^{-2}, 6 \times 10^{-2}, 1 \times 10^{-1}, 5 \times 10^{-1}\}$. We search for weight decay from $\{1 \times 10^{-5}, 5 \times 10^{-5}, 1 \times 10^{-4}, 5 \times 10^{-4}, 1 \times 10^{-3}, 5 \times 10^{-3}\}$. We search for the hidden dimension from $\{32, 64, 128, 256, 512\}$. The dropout rate is selected from $\{0.3, 0.4, 0.5, 0.6, 0.7\}$. Values leading to the lowest validation loss are selected for each dataset. Selected values of learning rate, weight decay, hidden dimension, and dropout rate on different datasets are shown in Table 7. All experiments in this paper are performed on a single NVIDIA Tesla A100 80G GPU.

### B.3  ATTACK SETTINGS

**Non-targeted Adversarial Attacks (Metattack) (Zügner & Günnemann, 2019).** We first evaluate the robustness of our method against the non-targeted adversarial attack method Metattack.

Table 7: Selected values of learning rate, weight decay, hidden dimension, and dropout rate.

| Hyper-parameters | Cora | Citeseer | PubMed | Polblogs |
|---|---|---|---|---|
| Learning Rate | $3 \times 10^{-2}$ | $3 \times 10^{-2}$ | $1 \times 10^{-3}$ | $6 \times 10^{-2}$ |
| Weight Decay | $5 \times 10^{-4}$ | $5 \times 10^{-4}$ | $5 \times 10^{-4}$ | $5 \times 10^{-5}$ |
| Hidden Dimension | 96 | 128 | 64 | 128 |
| Dropout Rate | 0.5 | 0.5 | 0.7 | 0.4 |

Metattack treats the graph as a hyperparameter to optimize and uses meta-gradients to solve the bilevel optimization problem, which minimizes the accuracy of node classification. We follow the implementation in (Zügner & Günnemann, 2019). As Metattack has several variants, we follow (Jin et al., 2020) and adopt the most destructive attack version, Meta-Self, on Cora, Citeseer, and Polblogs datasets. On Pubmed, we adopt the approximate version of Meta-Self, A-Meta-Self, to avoid memory and time overhead following the settings in (Jin et al., 2020). We measure the strength of the attack by the perturbation rate, which is the ratio of perturbed edges among all the edges in the graph. We evaluate our method and all baselines with perturbation rates ranging from $0$ to $25\%$ with a step of $5\%$.

**Targeted Adversarial Attack (Nettack) (Zügner et al., 2018).** We adopt Nettack as the targeted attack method in evaluating the robustness of our method. Nettack manipulates the graph structure and node features to degrade the classification accuracy on target nodes while minimizing the change in the graph's degree distribution and feature co-occurrences. We use the default attack settings in the original implementation in (Zügner et al., 2018). The nodes in the test set whose degree is larger than 10 are set as target nodes in the attack. In Nettack, the number of perturbations made on every targeted node is defined as the attack budget. Following (Jin et al., 2020), we evaluate our method and all baselines with attack budgets ranging from $1$ to $5$ with a step size of $1$. Following the settings in (Jin et al., 2020), we only sample $10\%$ of the target nodes for the evaluation on Nettack.

## C    More Experiment Results

### C.1    Study on the Effectiveness of Global Node Correlation Learning in RGA-IB

Our proposed RGA-IB learns node representations with dense graph attention to capture the correlations between all pairs of nodes in the graph for reducing the IB loss, which is not limited by the local dependency assumption enforced in GIB (Wu et al., 2020). To verify the effectiveness of global node correlation learning in RGA-IB, we compare RGA-IB with ablation models, named RGA-IB$_{\text{local}}$, that only capture local node correlations. Each attention weight matrix $B^{(\ell)}$ in the RGA-IB$_{\text{local}}$ models is replaced with $\widehat{B}^{(\ell)} = B^{(\ell)} \circ \text{sgn}\left(\sum_{l=1}^{L} A^l\right)$, where function $\text{sgn}(\cdot)$ applies element-wise to its input matrix, returning $1$ for elements greater than $0$ and returning $0$ for elements equal to $0$. As a result, only the attention weights between nodes that are connected within $L$-hops will be considered by $\widehat{B}^{(\ell)}$. Then, a two-layer RGA-IB only considers $2L$-hops local graph structure. We perform the ablation study for $L \in \{1, 2, 4, 8, 16\}$ on Cora, Citeseer, and Pubmed under Metattack with a perturbation rate of $25\%$. It is observed from the results in Table 8 that RGA-IB with more dense attention usually achieves better performance. For example, RGA-IB$_{\text{local}}$ with $L = 16$ outperforms RGA-IB$_{\text{local}}$ with $L = 1$ by $1.64\%$ on Cora. In addition, RGA-IB$_{\text{local}}$ models with more dense attention also features lower IB loss, showing that the dense graph attention better adheres to the IB principle, such that the learned node representations are less related to the input graph data while being more correlated to the class labels.

### C.2    Study on the RGA-IB attention graph

To study the effectiveness of the dense graph attention in RGA-IB on reducing the propagation of adversarial information in the attacked graph, we compare the attacked graph with the RGA-IB attention graph, which is created by connecting only pairs of nodes whose attention weights are larger than $0.2$. Nettack with an attack budget of $5$ is adopted for this experiment. We compare the frequency and cumulative frequency of the number of adversarial neighbors in the attacked graph and

Table 8: Ablation study on the effectiveness of global node correlation learning in RGA-IB. The study is performed on Cora, Citeseer, and Pubmed under Metattack with a perturbation rate of $25\%$.

| Datasets | | RGA-IB$_{local}$ | | | | | RGA-IB |
|---|---|---|---|---|---|---|---|
| | | $L=1$ | $L=2$ | $L=4$ | $L=8$ | $L=16$ | |
| Cora | IB Loss | $-0.160$ | $-0.175$ | $-0.180$ | $-0.196$ | $\underline{-0.205}$ | **-0.214** |
| | ACC | $69.82$ | $70.03$ | $71.32$ | $71.39$ | $\underline{71.42}$ | **71.43** |
| Citeseer | IB Loss | $-0.158$ | $-0.164$ | $-0.179$ | $-0.183$ | $\underline{-0.188}$ | **-0.189** |
| | ACC | $68.84$ | $69.14$ | $70.55$ | $70.89$ | $\underline{70.91}$ | **70.94** |
| Pubmed | IB Loss | $-0.167$ | $-0.175$ | $-0.187$ | $-0.190$ | $\underline{-0.192}$ | **-0.197** |
| | ACC | $85.55$ | $86.48$ | $87.32$ | $87.58$ | $\underline{87.84}$ | **87.92** |

the RGA-IB attention graph. The number of adversarial neighbors of a node in the attacked graph and the RGA-IB attention graph is computed by counting the number of nodes which have been altered by the Nettack within the two-hop neighborhood of that node. The frequency at a particular number of adversarial neighbors $p$ is the number of nodes which have $p$ adversarial neighbors with a two-hop neighborhood. The cumulative frequency at a particular number of adversarial neighbors $p$ is the fraction of the nodes which have $p$ or less adversarial neighbors in a two-hop neighborhood. The frequency and the cumulative frequency are illustrated in blue for the attacked graph and in red for the RGA-IB attention graph. The comparisons on Pubmed and Polblogs are illustrated in Figure 3.

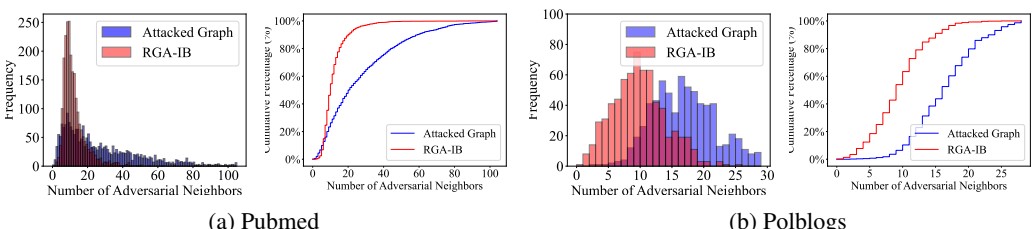

(a) Pubmed  (b) Polblogs

Figure 3: Comparisons on the frequency and cumulative frequency of the number of adversarial neighbors in the attacked graph and the RGA-IB attention graph for Pubmed and Polblogs. Nettack with an attack budget of 5 is adopted for this experiment. Only adversarial neighbors, which are perturbed nodes within two hops of a node, are counted as existing works GIB (Wu et al., 2020), RG-GIB (Dai et al., 2023b), and UGRL (Wang et al., 2023b) are limited by the two-hop local dependency assumption.

