# OpenReview forum: "Robust Graph Attention for Graph Adversarial Attacks: An Information Bottleneck Inspired Approach"
_ICLR.cc/2025/Conference — ICLR 2025 Conference Withdrawn Submission_

### Official Review · Reviewer_N3pK · 2024-10-26

**Soundness:** 2
**Presentation:** 2
**Contribution:** 2
**Rating:** 3
**Confidence:** 3

**Summary:**

This paper addresses the vulnerability of Graph Neural Networks (GNNs) to adversarial attacks and highlights that existing robust graph attention methods overlook the relationship between robustness and adherence to the Information Bottleneck (IB) principle. The study finds that attention-based GNNs with lower IB loss are more robust against adversarial attacks. Based on this insight, the authors propose a novel method, Robust Graph Attention inspired by Information Bottleneck (RGA-IB), which minimizes IB loss in GNNs. Experimental results show that GNNs with RGA-IB achieve better node classification accuracy under adversarial attacks compared to existing methods.

**Strengths:**

1. This work connects the Information Bottleneck with the robustness of graph attention methods and proposes a novel Robust Graph Attention method. The experiments also provide evidence of a connection between lower IB loss and improved robustness.

2. This work explains the differences between RGA-IB and other IB-based methods, highlighting that RGA-IB can capture global correlations, whereas others, like GIB, are limited by the local dependency assumption.

3. The experiments include several datasets, attacks, and baselines.

**Weaknesses:**

1. Limited Novelty: The model architecture is inspired by the Information Bottleneck, which I believe is similar in motivation to other IB-based models and not sufficiently novel. The only contribution of this method is actually the gradient decent of attention matrix B in Theorem 3.1.
Although the authors highlight differences between RGA-IB and other IB-based models, the distinction between local and global dependencies contributing to improved robustness is not entirely convincing. The robustness improvement might result from implementation details and hyperparameter tuning.

2. Evaluation: The experiments are not comprehensive enough.

* Baselines: How does the model compare to RUNG [1], ElasticGNN [2], SoftMedian [3], and TWIRLS [4]?
* Datasets: How does the model perform on heterophilic datasets?
* Attacks: How does the model perform under gradient-based local attacks and injection attacks?

[1] Robust Graph Neural Networks via Unbiased Aggregation

[2] Elastic Graph Neural Networks

[3] Robustness of Graph Neural Networks at Scale

[4] Graph Neural Networks Inspired by Classical Iterative Algorithms

3. The evidence of the connection between lower IB loss and improved robustness needs further development. Since GIB, UGRL, and RG-GIB are all IB-based models, why is RGA-IB more effective in reducing IB loss? Additionally, in Table 5, the authors should include these baselines to demonstrate that RGA-IB is more effective in reducing IB loss.

4. Minor suggestions: The presentation could be further enhanced. The method section appears somewhat disorganized, and the detailed computation of IB loss should be the main focus of this section. Equation (1) represents the gradient descent of matrix B and should not be formatted as a theorem. Additionally, Algorithm 1 lacks informativeness yet occupies excessive space. The experiment table's font is too small, making it inconsistent with the main text. In the experiment section, the authors should use a dedicated subsection for experimental settings to clearly present datasets, baselines, implementation details, and attacks, rather than introducing these aspects randomly within the main discussion.

**Questions:**

Refer to weaknesses

---

### Official Review · Reviewer_gmDJ · 2024-10-28

**Soundness:** 3
**Presentation:** 3
**Contribution:** 3
**Rating:** 6
**Confidence:** 3

**Summary:**

This paper introduces a novel graph attention method, Robust Graph Attention inspired by Information Bottleneck (RGA-IB) to enhance the robustness of Graph Neural Networks (GNNs) against adversarial attacks. By leveraging the Information Bottleneck (IB) principle, RGA-IB specifically aims to minimize IB loss in GNNs. This can reduce the mutual information between node representations and input features, while maximizing the mutual information between node representations and the desired outputs, and greatly increase the robustness of the GNN. Extensive experiments demonstrate that RGA-IB significantly improves node classification accuracy under various adversarial attacks on graphs compared to existing methods, effectively mitigating the impact of adversarial perturbations.

**Strengths:**

1. This paper is well written and effectively conveys the intended meaning.

2. The paper introduces a novel method, RGA-IB, which leverages the Information Bottleneck principle to enhance the robustness of GNNs against adversarial attacks. This is an innovative idea.

3. This method can be easily applied to different Graph Neural Networks, making it suitable for enhancing the robustness of existing models in practical applications.

4. The experiments are well-designed. Experimental results with error analysis prove that RGA-IB outperforms existing methods in terms of node classification accuracy under various adversarial attack scenarios.

5. The paper discusses the theoretical motivation behind using the IB principle for improving robustness. The mathematical derivation provides a clear relationship between reduced IB loss and improved adversarial robustness. It is a meaningful contribution towards understanding GNN’s robustness under adversarial attack.

**Weaknesses:**

1. The paper lacks details on computational resources and does not include comparisons related to memory usage or runtime, which are essential metrics for a comprehensive evaluation.

2. From an attacker’s perspective, the attacker may propose an adaptive attack against your proposed method. But this paper lacks exploration in this area.

3. This paper lacks comprehensive ablation studies. For example, Section 4.1 describes experimental settings such as the proportion of nodes used for training and the number of training epochs, which could also benefit from detailed ablation studies to better understand their impact on the model's performance.

4. The method introduces additional complexity in calculating attention weights and might cause higher computational costs, especially on large graphs. So perhaps this method is not feasible in real world applications.

**Questions:**

1. Your paper introduces a novel approach by minimizing the IB loss in GNNs. But can you discuss the trade-off in your method between increased training complexity and enhanced robustness by doing more experiments or from a mathematical perspective?

2.  Will this method lead to a decrease in accuracy in non-adversarial settings compared with the baselines?

---

### Official Review · Reviewer_e6KR · 2024-10-29

**Soundness:** 2
**Presentation:** 3
**Contribution:** 2
**Rating:** 3
**Confidence:** 4

**Summary:**

This paper introduces a new attention-based graph neural network named RGA-IB to enhance GNN's robustness. The method is inspired by Information Bottleneck (IB) and aims to make GNNs more robust by minimizing the IB loss. The experimental results show the effectiveness of the proposed method.

**Strengths:**

This paper has the following Strengths:
- This paper is well-written and easy to understand.
- The phoneme found in this paper that the IB loss is related to the robustness of attention-based graph neural networks is interesting.

**Weaknesses:**

This paper has the following Weaknesses:
- This research focuses on attention-based graph neural networks which might be a little narrow.
- The attack methods used in this paper are old. For the untargeted attack, the authors only used meta attack which was published in 2019. For the targeted attack, the authors only used a method that was published in 2018. I strongly suggest the authors add more and also some newer attack methods. For example, for untargeted attacks, the authors can also use this method [1]. For untargeted attacks, the authors can also use these methods [2,3]. Only one old attack method for each category is not convincing enough to prove the robustness is related to IB loss.
- The scalability of the proposed method is not clear. What about the running time of RGA-IB and GPU memory needed compared with other robust GNNs used in the paper such as Pro-GNN and Difformer? It is recommended to have a part to discuss this. The datasets used are relatively small. The authors can consider conducting some experiments on large-scale datasets such as OGB-arxiv [4].
-------------------
**References**
[1] Scalable attack on graph data by injecting vicious nodes. *Data Min Knowl Disc* 34, 1363–1389 (2020).
[2] Adversarial Attacks on Graph Neural Networks via Node Injections: A Hierarchical Reinforcement Learning Approach. In *WWW* (2020).
[3] Robustness of Graph Neural Networks at Scale. In *NeurIPS* (2021).
[4] Open graph benchmark: Datasets for machine learning on graphs. In *NeurIPS*.

**Questions:**

I also have a few questions
- Could authors try to specify the environment in the readme file in the anonymous GitHub Repo? I tried to run it but the results are not as good as described in the paper. I also suggest the authors describe the computational devices used in the paper to enhance reproducibility.
- For other questions, please refer to the weakness part.

---

### Official Review · Reviewer_2vDN · 2024-10-30

**Soundness:** 3
**Presentation:** 3
**Contribution:** 2
**Rating:** 3
**Confidence:** 3

**Summary:**

This paper proposes Robust Graph Attention inspired by Information Bottleneck (RGA-IB), a novel attention mechanism for GNNs aimed at enhancing robustness against adversarial attacks. By minimizing IB loss, RGA-IB aligns node representations more closely with target outputs, reducing vulnerability to noisy or adversarial inputs. Experiments show that GNNs equipped with RGA-IB outperform other robust GNNs, achieving higher accuracy and robustness in semi-supervised node classification under adversarial conditions.

**Strengths:**

1. The paper is well-written and easy to follow.
2. The proposed framework is supported by theoretical analysis.
3. The experiments are comprehensive.

**Weaknesses:**

1. The major concern with this framework is its time complexity; however, there is no time complexity analysis throughout the paper.
2. The experiments mainly focus on small graph datasets. I would suggest the authors conduct more experiments on larger datasets.
3. Intuitively, lower IB loss would lead to lower accuracy on a clean graph dataset. However, as I noticed in Tables 1, 2, and 3, RGA-IB consistently achieves the best result in the clean dataset setting. Could the authors provide an explanation for this phenomenon?
4. I would suggest the authors add more detailed descriptions of how RGA-IB differs from previous work to further highlight its contributions, especially in relation to studies that discuss Information Bottleneck theory [1].

[1] A Unified Framework of Graph Information Bottleneck for Robustness and Membership Privacy. KDD 2023.

**Questions:**

Please refer to weakness.

---

### Official Review · Reviewer_4n9H · 2024-11-03

**Soundness:** 2
**Presentation:** 2
**Contribution:** 2
**Rating:** 3
**Confidence:** 4

**Summary:**

The paper introduces Robust Graph Attention inspired by Information Bottleneck (RGA-IB), a method designed to improve the robustness of Graph Neural Networks (GNNs) against adversarial attacks by leveraging the Information Bottleneck (IB) principle. RGA-IB minimizes IB loss to enhance GNNs' resilience, providing a new perspective on attention-based GNNs by explicitly incorporating IB in the attention mechanism. The paper demonstrates the superiority of RGA-IB in semi-supervised node classification under various adversarial attack scenarios, outperforming existing methods in terms of node classification accuracy on public benchmarks.

**Strengths:**

1.	Comprehensive Empirical Validation: The method is thoroughly tested on multiple public datasets (e.g., Cora, Citeseer, Pubmed, Polblogs), with results indicating superior performance in semi-supervised node classification under various adversarial attacks. The experiments demonstrate the method’s effectiveness and generalizability.
2.	Enhanced Robustness through IB Principle: This paper connects the Information Bottleneck (IB) principle and improved robustness, and it provides empirical experiments to validate this strong connection.
3.	Clear Explanation of Proposed Method: This paper provides a detailed and clear explanation of the proposed architecture and technique.

**Weaknesses:**

1.	Scalability and Computational Complexity: The proposed method introduces an attention matrix BBB of size N×NN \times NN×N, which leads to significant computational and memory demands, especially during gradient computation and matrix updates. This could limit the scalability of the method, particularly on large-scale graphs, and a complexity analysis is needed to better understand these demands.
2.	Evaluation on Small to Medium Datasets: The datasets used for evaluation, such as Polblogs and Pubmed, are not large-scale. It is unclear how well the proposed method would generalize to or perform on larger graphs, raising questions about its applicability to real-world scenarios involving massive networks.
3.	Limited Novelty: While the method applies the Information Bottleneck (IB) principle to improve robustness, this concept has already been explored in other works for enhancing robustness. Consequently, the contribution and novelty of this specific architecture may be limited compared to existing research.
4.	Presentation and Organization Issues: Although the paper provides a detailed explanation of the method, there is considerable room for improvement in presentation, particularly in the methods section. The implementation details should not dominate the main part of the methodology, and the theoretical justification is lacking. The theorem and lemma only explain how to derive the gradient descent formulation.

**Questions:**

Please refer to the weakness.

---

### Note · Authors · 2025-01-15

I have read and agree with the venue's withdrawal policy on behalf of myself and my co-authors.